# 🌀 GyroSwin: 5D Surrogates for Gyrokinetic Plasma Turbulence Simulations

**Fabian Paischer** [* 1,3]   **Gianluca Galletti** [* 1]   **William Hornsby**[2]   **Paul Setinek**[1]

**Lorenzo Zanisi**[2]   **Naomi Carey**[2]   **Stanislas Pamela**[2]   **Johannes Brandstetter**[1,3]

[1] ELLIS Unit, Institute for Machine Learning, JKU Linz
[2] United Kingdom Atomic Energy Authority, Culham campus
[3] EMMI AI, Linz
{paischer,galletti,brandstetter}@ml.jku.at

⬡ ml-jku/neural-gyrokinetics,   🤗 ml-jku/gyroswin

## Abstract

Nuclear fusion plays a pivotal role in the quest for reliable and sustainable energy production. A major roadblock to viable fusion power is understanding plasma turbulence, which significantly impairs plasma confinement, and is vital for next-generation reactor design. Plasma turbulence is governed by the nonlinear gyrokinetic equation, which evolves a 5D distribution function over time. Due to its high computational cost, reduced-order models are often employed in practice to approximate turbulent transport of energy. However, they omit nonlinear effects unique to the full 5D dynamics. To tackle this, we introduce GyroSwin, the first scalable 5D neural surrogate that can model 5D nonlinear gyrokinetic simulations, thereby capturing the physical phenomena neglected by reduced models, while providing accurate estimates of turbulent heat transport. GyroSwin (i) extends hierarchical Vision Transformers to 5D, (ii) introduces cross-attention and integration modules for latent 3D↔5D interactions between electrostatic potential fields and the distribution function, and (iii) performs channelwise mode separation inspired by nonlinear physics. We demonstrate that GyroSwin outperforms widely used reduced numerics on heat flux prediction, captures the turbulent energy cascade, and reduces the cost of fully resolved nonlinear gyrokinetics by three orders of magnitude while remaining physically verifiable. GyroSwin shows promising scaling laws, tested up to one billion parameters, paving the way for scalable neural surrogates for gyrokinetic simulations of plasma turbulence.

## 1   Introduction

Nuclear fusion promises sustainable energy by fusing hydrogen isotopes within an ionized plasma. Because the plasma reaches temperatures of hundreds of millions of degrees, it must be confined by a magnetic field. During confinement, turbulence can arise from micro-instabilities, leading to energy and particle transport toward the reactor walls. This causes plasma to leak from its magnetic cage, resulting in heat and density loss, and therefore impaired energy production. The design and control of plasma scenarios strictly require knowledge of turbulent transport, which can be obtained via nonlinear gyrokinetic simulations that evolve a 5D Partial Differential Equation (PDE) over time.

The computational cost of nonlinear gyrokinetic simulations is prohibitive. Therefore, QuasiLinear approaches (QL), such as QuaLiKiz (Bourdelle et al., 2015; Citrin et al., 2017) and TGLF (Staebler

---

*Equal contribution

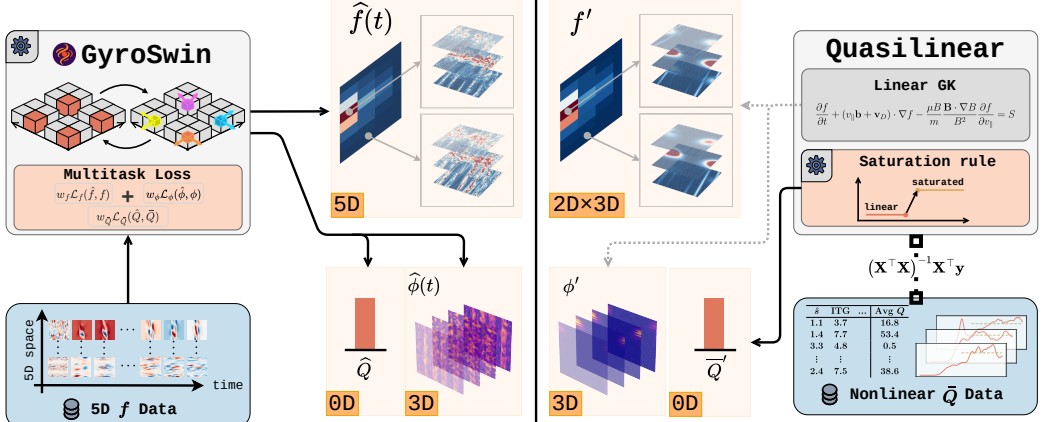

Figure 1: **Left:** GyroSwin models the 5D distribution function of nonlinear gyrokinetics and incorporates integration blocks to predict 3D electrostatic potential fields and scalar heat flux. **Right:** ROMs (quasilinear) solve a cartesian product of 2D modes in spectral space and 3D fields. Furthermore, they rely on saturation rules to approximate the nonlinear flux spectrum.

et al., 2007; Staebler & Kinsey, 2010), are commonly used to approximate turbulent transport. They are based on a Reduced-Order Model (ROM) that operates in 3D and adopt a so-called saturation rule to estimate nonlinear fluxes. These saturation rules usually employ free parameters that are fitted to nonlinear flux data. However, QL models neglect essential parts of the nonlinear physics that contribute substantially to evolving turbulent transport, namely zonal flows. Therefore, a reliable estimate of turbulent fluxes to date is only attainable via expensive nonlinear gyrokinetic simulations.

To tackle the prohibitive cost of nonlinear gyrokinetic simualtions, we introduce GyroSwin, a scalable neural surrogate model to efficiently approximate turbulent transport. GyroSwin is based on three essential ingredients: (i) extension of a Swin transformer (Liu et al., 2021) to 5D data, (ii) ilatent cross-attention and integration modules for interaction between 5D and 3D fields, as well as 5D → 3D integration, and (iii) channelwise mode separation informed by nonlinear physics. To ground GyroSwin on physically meaningful quantities, we train it in a multitask manner on the 5D distribution function and derived quantities thereof, such as 3D potential fields, and scalar fluxes.

GyroSwin accurately captures nonlinear physics of 5D gyrokinetics. To verify this, we infer physical downstream quantities of the predicted 5D PDE, such as heat flux, turbulence related spectra, and zonal flows. We find that the predicted quantities align well with the ground truth for unseen simulations and GyroSwin scales favorably compared to other neural surrogate approaches. Furthermore, GyroSwin is three orders of magnitude faster than the numerical code GKW for the adiabatic electron approximation. In summary, our contributions are as follows.

- We introduce GyroSwin, a multitask hierarchical Vision Transformer that handles data of arbitrary dimensionality. It is trained on the adiabatic electron approximation using channelwise mode separation to predict the 5D PDE, 3D electrostatic potentials, and scalar fluxes simultaneously.

- We propose latent cross-attention between fields of varying dimensionality, as well as integration layers that enable integrating along various dimensions of the 5D field.

- We demonstrate that GyroSwin captures nonlinear dynamics in the 5D field, improves time-averaged flux predictions over state-of-the-art ROMs, and significantly outperforms deep learning surrogates on plasma turbulence modelling.

- We demonstrate promising scaling laws of GyroSwin, tested up to 1B parameters.

## 2 Background and Related Work

**5D Gyrokinetics.** To determine turbulent transport, one must simulate the evolution of electrons and ions in the plasma over time. In principle, this could be achieved by modelling each species as a particle in the plasma (Birdsall & Langdon, 2005; Tskhakaya, 2008). However, because the number

of particles in a plasma can exceed $10^{20}$, this approach is computationally expensive. An alternative approach is to statistically model ensembles of particles in the plasma using a distribution function. This approach is termed plasma kinetics and is desirable to analyse stability in a bulk of plasma.

Plasma kinetics models the time evolution of electrons and ions via the distribution function $f$ based on 3D coordinates, their parallel and perpendicular velocities, and the angle around the field lines. Gyrokinetics (Krommes, 2012) is a reduced form of plasma kinetics that is more efficient by averaging out the angle around the field lines and only considering a particle's guiding center. Local gyrokinetics focuses on perpendicular spatial scales comparable to the gyroradius and on frequencies much lower than the particle cyclotron frequency, the circular motion of charged particles around magnetic field lines due to the Lorentz force. Hence, the 5D gyrokinetic distribution function can be written as $f = f(k_x, k_y, s, v_\parallel, \mu)$, where $k_x$ and $k_y$ are spectral coordinates (for the spatial $x$ and $y$), $s$ is the toroidal coordinate along a field line, and $v_\parallel$ and $\mu$ represent parallel and perpendicular velocity components, respectively. The perturbed time-evolution of $f$, for each species (ions and electrons), is governed by

$$\underbrace{\frac{\partial f}{\partial t} + (v_\parallel \boldsymbol{b} + \boldsymbol{v}_D) \cdot \nabla f - \frac{\mu B}{m} \frac{\boldsymbol{B} \cdot \nabla B}{B^2} \frac{\partial f}{\partial v_\parallel}}_{\text{Linear}} + \underbrace{\boldsymbol{v}_\chi \cdot \nabla f}_{\text{Nonlinear}} = S \, , \tag{1}$$

where $v_\parallel \boldsymbol{b}$ is the motion along magnetic field lines, $\boldsymbol{b} = \boldsymbol{B}/B$ is the unit vector along the magnetic field $\boldsymbol{B}$ with magnitude $B$[2], $\boldsymbol{v}_D$ the magnetic drift due to gradients and curvature in $\boldsymbol{B}$, and $\boldsymbol{v}_\chi$ describes drifts arising from the $\boldsymbol{E} \times \boldsymbol{B}$ force, a key driver of plasma turbulence. The nonlinear term models the interaction between the distribution function $f$ and the electrostatic potential, $-\nabla \phi = \boldsymbol{E}$, which comes from the velocity space integral of $f$ itself, and describes turbulent advection. The resulting nonlinear coupling constitutes the computationally most expensive term. Finally, $S$ is the source term that represents collisions between particles. For a more detailed derivation of the gyrokinetic equation, see Appendix A.

**Quantities of interest.** In practice, the main quantity of interest is the radial transport of energy (heat) towards the reactor wall, which is crucial for reactor design. It can be obtained by integrals on the 5D distribution function as

$$Q = \int \mathbf{C} \int \mathbf{v}^2 \boldsymbol{\phi} \boldsymbol{f} \, \mathrm{d}v_\parallel \mathrm{d}\mu \, \mathrm{d}x \mathrm{d}y \mathrm{d}s, \qquad \phi = \mathbf{A} \int \mathbf{J_0} \boldsymbol{f} \, \mathrm{d}v_\parallel \mathrm{d}\mu, \tag{2}$$

where $\mathbf{A}, \mathbf{C} \in \mathbb{R}^{x \times s \times y}$ comprise geometric and operating parameters, $\mathbf{v} \in \mathbb{R}^{v_\parallel \times \mu}$ denotes particle energy, and $\mathbf{J_0}$ denotes the zeroth-order Bessel function and $\phi \in \mathbb{R}^{x \times s \times y}$ is the electrostatic potential. In a tokamak, $\boldsymbol{B}$ largely points in the toroidal direction, $k_x$ encodes the radial direction, and $k_y$ is perpendicular to $k_x$ (binormal $\approx$ poloidal) in Fourier space. Electrostatic fluctuations that drive turbulence occur mainly in the $k_y$ direction, while $k_x$ encodes the radial structural properties of turbulence, i.e. the size of emerging eddies.

Radial turbulent transport occurs only if the amplitude of the $k_y$ modes is non-zero, as this results in $\boldsymbol{E} \times \boldsymbol{B}$ drift, and consequently in fully developed nonlinear turbulence. A turbulent simulation usually follows a certain pattern. In the linear phase, different modes in $k_y$ start growing due to underlying micro-instability, resulting in an initial increase in heat flux. Afterwards, the simulations enter the nonlinear (saturated) regime where modes start interacting and the system converges into a statistically steady state. This state is controlled by zonal flows that shear and break up arising eddies, effectively dampening turbulence (Itoh et al., 2006).

To analyse turbulence, practitioners usually investigate various spectra along the $k_y$ direction, as those provide insights into turbulent transport. Most importantly, $Q(k_y)$ provides insight into which modes dominate the turbulence energy:

$$Q(k_y) = \sum_{v_\parallel, \mu, s, k_x} \boldsymbol{Q}(v_\parallel . \mu, s, k_x, k_y) \, , \tag{3}$$

where $\boldsymbol{Q} \in \mathbb{R}^{v_\parallel \times \mu \times s \times k_x \times k_y}$ is the flux field, which aggregates to the scalar flux $Q$ in Equation (2). In addition $W(k_y) \in \mathbb{R}^{k_y}$ describes the intensity of turbulence along $y$, and zonal flows dampen

---

[2]We adopt uppercase notation for vector fields $\boldsymbol{E}$ and $\boldsymbol{B}$ to adhere with literature.

turbulence resulting in the statistically steady state which can be isolated as the first mode of $\phi$ in $k_y$ direction:

$$W(k_y) = \sum_{s,k_x} |\phi(k_x, s, k_y)|^2 \,, \quad \phi_{\text{ZF}}(x, t) = \sum_{k_x} \phi_{k_x, \, k_y=0}(t) \, e^{ik_x x}. \tag{4}$$

Since the system converges to a statistically steady state, practitioners usually investigate time-averaged quantities of $Q(k_y)$, $W(k_y)$, $\phi_{\text{ZF}}$, and scalar heat flux trace $\bar{Q}$.

**State-of-the-art numerical approximations.** Solving Equation (1) numerically is computationally expensive, especially for high-fidelity simulations across ion and electron scales. Quasilinear (QL) models mitigate the main source of computational cost and are commonly adopted in integrated modelling pipelines (Mulders et al., 2021; Citrin et al., 2024). They are based on linear simulations that neglect the nonlinear term in Equation (1) and solve for multiple $f$ for each mode in $k_x \times k_y$ independently. This results in a speedup as the order of $f$ is reduced to $\hat{f} = f(v_{||}, \mu, s)$. As a result, linear simulations do not account for interaction between modes, but only compute linear growth rates and potential transport contributions for each of them. This mode-wise transport contribution is combined via so-called *saturation rules* by

$$\bar{Q} = \int Q_{\text{QL}}(k_y) \hat{W}(k_y) dk_y \,, \tag{5}$$

where $Q_{\text{QL}}(k_y)$ is the flux contribution per mode, and $\hat{W}(k_y)$ is a modelled weighting function which approximates $W(k_y)$ and whose free parameters are fit to nonlinear simulations. Since QL models are based on linear simulations, they neglect nonlinear physics that substantially impact turbulence. Another limitation of QL models are operating parameter regions leading to strong turbulence (Staebler et al., 2024; Kiefer et al., 2021; Dimits et al., 2000b; Bourdelle et al., 2008).

**Machine learning for Gyrokinetics.** Machine Learning offers a fruitful alternative to ROMs. Most literature to date has focused on multilayer perceptrons (van de Plassche et al., 2020; Citrin et al., 2023; Zanisi et al., 2024, MLP) or gaussian processes (Hornsby et al., 2024). They usually map from operating parameters, such as temperature or density gradients to $\bar{Q}$ predicted via ROMs. Hence, they are inherently limited by the capabilities of ROMs used to produce the training data. Few works have attempted to train machine learning surrogates for nonlinear gyrokinetics. Narita et al. (2022) leverage convolutional neural networks on 2D wavenumber ($k_x \times k_y$) slices to predict time to saturation and heat flux. Building on this idea, Honda et al. (2023) include snapshots of electrostatic potentials as additional channels. More recently, Wan et al. (2025) investigate transfer from low-fidelity to higher-fidelity simulations in a reduced 1D space. Our proposed GyroSwin is fundamentally different, as it is trained directly on the 5D distribution function of nonlinear gyrokinetics.

**Neural surrogates.** Over recent years, deep neural network-based surrogates have emerged as a computationally efficient alternative in science and engineering (Thuerey et al., 2021; Zhang et al., 2023; Brunton et al., 2020), impacting weather forecasting (Kurth et al., 2023; Bi et al., 2023; Lam et al., 2023; Nguyen et al., 2023; Bodnar et al., 2024), protein folding (Jumper et al., 2021; Abramson et al., 2024), material design (Merchant et al., 2023; Zeni et al., 2025; Yang et al., 2024), and multi-physics modelling Alkin et al. (2024b). These success stories share the common thread of deep learning surrogates overcoming seemingly insurmountable challenges (Brandstetter, 2024). Especially for weather modelling, Vision Transformer (Dosovitskiy et al., 2021, ViT) and especially Swin Transformer (Liu et al., 2021) have shown exceptional performance, which manifested in the first medium-range weather modelling surpassing numerical accuracy (Bi et al., 2023), and the current state-of-the-art foundation model of the atmosphere(Bodnar et al., 2024). In gyrokinetics we face similar challenges in terms of fidelity, complexity and local couplings. However, these challenges are even exacerbated for gyrokinetics due to its 5D nature.

## 3  GyroSwin

State-of-the-art ROMs neglect 5D physical phenomena that are crucial for reliable turbulent transport estimates. To remedy this, we develop a neural surrogate, GyroSwin, that directly learns to evolve the 5D distribution function of nonlinear gyrokinetics over time. The most important aspect

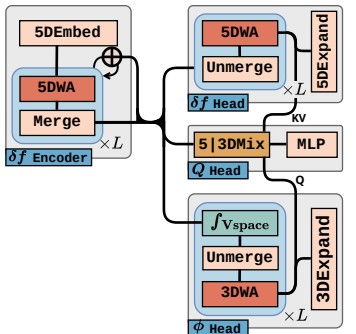
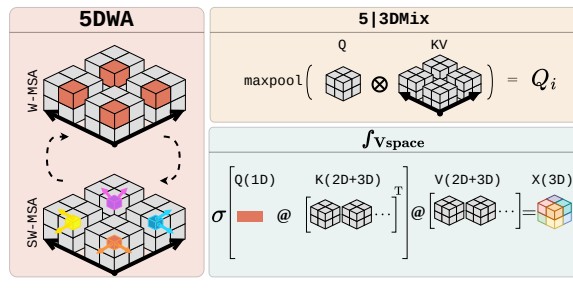

(a) High-level GyroSwin architecture.     (b) 5D attention, latent mixing and integrator layers.

Figure 2: **Left:** GyroSwin receives the 5D distribution function as input and predicts the evolved 5D distribution function, as well as the respective 3D potential and heat flux at the next timestep. **Right:** Essential building blocks and integrator layers that enable multitask training. The latent 5D space is integrated over velocity space to obtain a latent 3D field for potential prediction via cross-attention.

in designing such a surrogate model is scalability as nonlinear gyrokinetic equations can span multiple turbulence modalities for different species at varying resolutions. Furthermore, downstream physical quantities, such as electrostatic potential fields and scalar fluxes should be consistent with the predicted 5D field. Finally, understanding the structure of the 5D field enables us to bake inductive biases into the model architecture that improve capturing nonlinear dynamics, i.e., zonal flows. Based on these observations we pose the following desiderata for designing GyroSwin.

1. **Scalability.** We identify two main candidates: (1) Convolution-based (Fukushima, 1980; LeCun et al., 1989, CNN) or FNOs (Li et al., 2021), and (2) Vision Transformers (Dosovitskiy et al., 2021, ViT). To preserve locality of the 5D field, CNNs or FNOs require factorized implementations (Wang et al., 2017; Tran et al., 2023) that do not scale well (see Table 3). ViTs are in principle applicable to 5D, however flattening a 5D field results in extremely long patch sequences. Due to their quadratic complexity (Vaswani et al., 2017), ViTs do not scale well to high resolution input. Hierarchical processing with linear attention, as in Swin (Liu et al., 2021), provides an efficient way to preserve locality via local window attention and was successfully applied to 3D (Liu et al., 2022) or 4D (Kim et al., 2023) input. Finally, patch embeddings can be compressed to a lower resolution to facilitate scalability.

2. **Modelling latent integrals.** Electrostatic potentials and heat fluxes are computed as integrals (Equation (2)). We replicate this operation in the latent space of GyroSwin. To obtain the 3D latent fields, we introduce a latent integrator module that aggregates over the velocity space. Similarly, taking inspiration from Equation (2), the flux prediction is based on cross-attention pooling of 3D and 5D latents.

3. **Inductive biases.** The main benefit of GyroSwin over ROMs is that nonlinear dynamics can be captured that severely affect emerging turbulence. To bias GyroSwin towards nonlinear zonal flows, we perform *channelwise mode separation*. Specifically, we separate the zonal flow mode from Equation (4) from the other modes and transfer the spectral space to real space to retain the same dimensions. Then we add real and imaginary parts of the zonal flow as additional channel.

Taking into account these requirements, we design GyroSwin as a Swin-based UNet (Ronneberger et al., 2015) with multiple branches to accommodate multitask predictions, and physical priors. The following paragraphs explain the main architectural components of GyroSwin. We describe them in the domain-specific case of gyrokinetics with 5D shifted Window Attention (5DWA) and 5D up/downsampling layers, initially introduced by Galletti et al. (2025). All of them are generalized to n-dimensions through adaptive window partitioning and local convolution.

**5D shifted window attention.** The main characteristic of Swin Transformer (Liu et al., 2021) is local attention in fixed window sizes. This has a significant advantage over ViTs, reducing the quadratic complexity in token count to near-linear. More information on complexity for Swin and ViT can be found in Appendix H.1. The core component of GyroSwin is 5DWA (see Figure 2b). In 5DWA, attention is performed across tokens within 5D windows of size

$M := M_{v_\parallel} \times M_{v_\mu} \times M_s \times M_x \times M_y$. Let $\boldsymbol{X} \in \mathbb{R}^{b \times v_\parallel \times v_\mu \times s \times x \times y \times d}$ be a 5D field of batch size $b$ and hidden dimension $d$. After partitioning $\boldsymbol{X}$ into non-overlapping windows, we obtain $\boldsymbol{X}_w \in \mathbb{R}^{(b \cdot N_{win}) \times \left( M_{v_\parallel} \cdot M_{v_\mu} \cdot M_s \cdot M_x \cdot M_y \right) \times d}$, with $N_{win} = v_\parallel / M_{v_\parallel} \cdot v_\mu / M_{v_\mu} \dots$ denoting the number of windows. Self-attention is performed within each window in parallel across all windows. To enable interaction across neighboring windows we stack Window Multi-Head Attention (W-MSA) and Shifted Window MSA (SW-MSA) blocks. This results in approximate global attention with increasing depth. We define W-MSA for any n-dimensional $x$ as

$$\texttt{W-MSA}(\boldsymbol{X}) = \boldsymbol{W}^o \left[ \text{SoftMax} \left( \frac{(\boldsymbol{X}_w \boldsymbol{W}_h^Q)(\boldsymbol{X}_w \boldsymbol{W}_h^K)^\top}{\sqrt{d}} \right) (\boldsymbol{X}_w \boldsymbol{W}_h^V) \right]_{h \in \text{heads}}, \qquad (6)$$

where $\boldsymbol{W}_h^Q, \boldsymbol{W}_h^K, \boldsymbol{W}_h^V$ are the head-wise query, key and value projection matrices for head $h$, and $\boldsymbol{W}^o$ is the output matrix. The shifted version SW-MSA is defined as spatially shifting the window partition stencil prior to Equation (6), and reversing the shift after

$$\texttt{SW-MSA}(x) = \text{roll} \left( \texttt{W-MSA} \left( \text{roll} \left( \boldsymbol{X}, -\frac{M}{2} \right) \right), +\frac{M}{2} \right), \qquad (7)$$

where $\text{roll}\left( \boldsymbol{X}, \pm\frac{M}{2} \right)$ is a matrix roll operation, which circularly shifts each dimension of $x$ by $\pm\frac{M}{2}$ in the respective dimension. For details on batched implementation of the cyclic shift we refer to Liu et al. (2021) and Kim et al. (2023).

**Up/Downsampling.** Following (Dosovitskiy et al., 2021), we employ *patch embedding* to split the input into non-overlapping patches via 5D local convolution and embed them into tokens. The inputs are then spatially downsampled by the patch size and channels are projected to dimension $C$. The downsampling path of the UNet interleaves 5DWA blocks with *patch merging* layers to compress the 5D field at each stage. We extend these patch merging layers from Liu et al. (2021) to 5D: they concatenate features of $2 \times 2 \times 2 \times 2 \times 2$ neighboring patches and apply an MLP, reducing resolution by $2^5$ times. The output dimension is set to $2C$, which doubles the channels after each merge. We store intermediate feature maps for skip connections to produce hierarchical representations at different resolutions, resulting in growing receptive field and lower computational cost at lower stages. For upsampling we employ *patch expansion* layers, reversing the patch embedding (and merging) via a linear projection and rearrangement to the original field size. Figure 2a sketches the architecture of GyroSwin, incorporating patch embedding/merging/expansion blocks along with 5DWA.

**Multitask training.** To ensure that GyroSwin adheres to downstream integrated quantities, such as electrostatic potential fields $\phi$ and scalar flux $Q$, we add prediction tasks for each of them in addition to predicting $f$. The $\phi$-head is constructed of $L$ blocks, each of which employs an integral layer to reduce the 5D latent to a 3D latent followed by 3DWA layers and expansion layers until the original spatial resolution is recovered. For the $Q$-head, we perform max pooling over the integrated 3D space followed by an MLP head. Importantly, we also employ latent 5D $\leftrightarrow$ 3D mixing layers at each block to facilitate latent communication among the different heads. For multitask training of GyroSwin, we introduce loss terms for each head with weighting factors:

$$\mathcal{L} = w_f \mathcal{L}_f(\hat{f}, f) + w_\phi \mathcal{L}_\phi(\hat{\phi}, \phi) + w_{\bar{Q}} \mathcal{L}_{\bar{Q}}(\hat{Q}, \bar{Q}). \qquad (8)$$

**5D$\leftrightarrow$3D mixing and integrator modules.** Multitask training on 3D potential fields requires reducing the 5D latent space to a latent 3D space. To this end, we incorporate latent integrator modules (see $\int_{\textbf{Vspace}}$ block in Figure 2b). They perform cross-attention between a learnable 1D query $\boldsymbol{Q} \in \mathbb{R}^{1 \times d}$, which is broadcasted over the velocity space of $\boldsymbol{K}, \boldsymbol{V} \in \mathbb{R}^{(b \cdot s \cdot x \cdot y) \times v_\parallel \times \mu \times d}$. This procedure is reminiscent of perceiver-style pooling (Jaegle et al., 2021) with a single fixed query across velocity dimensions. Moreover, we allow interaction between 3D and 5D latents via cross-attention in each block $L$, where $\boldsymbol{Q} \in \mathbb{R}^{b \times s \times x \times y \times d}$ and $\boldsymbol{K}, \boldsymbol{V} \in \mathbb{R}^{b \times (v_\parallel \cdot v_\mu) \times s \times x \times y \times d}$. Figure 2b shows an example of latent 3D $\rightarrow$ 5D interaction. by swapping $\boldsymbol{Q}$ with $\boldsymbol{K}$ and $\boldsymbol{V}$, we also obtain 5D $\rightarrow$ 3D cross-attention. We stack both cross-attention layers to obtain our *5|3DMix* bloks.

## 4 Experiments

In this section we elaborate on our experimental setup, ranging from data generation to baselines and our evaluation setup. For implementation details see Appendix D.

**Data Generation.** We run nonlinear simulations using the numerical code GKW (Peeters et al., 2009), varying noise amplitude of initial conditions and four operating parameters, namely the safety factor $q$, the magnetic shear $\hat{s}$, the ion temperature gradient $R/L_t$ and the density gradient $R/L_n$. To reduce the computational burden of data generation, we consider the adiabatic electron approximation at a resolution of $(32 \times 8 \times 16 \times 85 \times 32)$, i.e. we only consider turbulence originating from ion temperature gradients. We use latin hypercube sampling to uniformly populate the parameter space (McKay et al., 1979) and chose parameter regions to ensure that the resulting simulations are highly turbulent. The distribution of operating parameters and corresponding heat flux can be observed in Figure 5. We run each simulation for a total of 31,920 steps, which are averaged every 40 steps and subsampled every third step, resulting in a total of 266 snapshots. Each snapshot comes with two channels, representing the real and imaginary parts of the ballooning transform, commonly used for plasma coordinates. We neglect the first 80 snapshots of each simulations as those correspond to the linear phase where turbulence is not fully established yet. The entire dataset comprises 255 simulations, based on which we assemble two training subsets, one comprising 48 simulations and another one comprising 241 simulations. We use the small subset for comparison to baselines and the latter for scaling experiments. In total, this results in 44,585 training samples, of which 8880 are used for the small subset. We provide visualizations of sample snapshots of the 5D fields in Appendix C.

**Baselines.** We implement three types of baselines: (**i**) As a state-of-the-art ROM, we implement the QuaLiKiz saturation rule (Bourdelle et al., 2015) and fit it to nonlinear fluxes of our training set, similarly to Kumar et al. (2021). For details, see Appendix B. (**ii**) Tabular regressors, such as Gaussian Process Regression (GPR) for nonlinear fluxes as proposed by (Hornsby et al., 2024), and a MLP trained in the same manner, akin to van de Plassche et al. (2020). (**iii**) Neural surrogates, trained to predict the 5D density function: Fourier Neural Operator (Li et al., 2021, FNO), PointNet (Qi et al., 2016), Transolver (Wu et al., 2024), and vanilla ViT (Dosovitskiy et al., 2021). Additional information on baselines can be found in Appendix D.1.

**Evaluation.** The promise of GyroSwin is that it can replace QL approaches as it is trained on the full 5D distribution function, but maintains efficiency and scalability. To properly evaluate whether our GyroSwin yields improvements over QL models, we compile a set of nonlinear simulations in a high ion temperature gradient regime, ensuring a strong turbulence regime. We set aside 14 of the 255 simulations that we generate in total to evaluate for in-distribution (**ID**) and out-of-distribution (**OOD**) generalization. For the **ID** set, we identify a region in the 4D parameter set that lives within the convex hull of the training set, but is unseen to the model. Conversely, we ensure that the **OOD** set is not governed by the convex hull of the training simulations. In total, we compile select six simulations for the **ID** set and five simulations for the **OOD** set. The remaining three simulations are used as a validation set during training. All of them are excluded from the training set.

## 5 Results

As shown in Table 1 different methods are restricted to certain evaluations, i.e. tabular regressors can only be evaluated on nonlinear flux prediction (scalars), while QL models can additionally be used to evaluate for diagnostics. Neural surrogates that predict the full 5D field, such as GyroSwin, are the only class that can also be evaluated for zonal flows. In line with this observation, we provide results for each of the three categories based on their capabilities.

Table 1: Comparison of different surrogate approaches by capabilities.

| Method | Average Flux | Diagnostics | Zonal Flows | Turbulence |
|---|---|---|---|---|
| Tabular Regressors, e.g., GPR, MLP | 1D→0D | ✗ | ✗ | ✗ |
| SOTA Reduced Numerical modelling, e.g., QL | 3D→0D | 3D→1D | ✗ | ✗ |
| Neural Surrogates, e.g. GyroSwin (Ours) | 5D→0D | 5D→1D | 5D→1D | 5D→5D |

**5D→5D Turbulence modelling.** To evaluate for 5D turbulence modelling capabilities, we perform an autoregressive rollouts with the neural surrogates and measure correlation time. Following Alkin et al. (2024a), we define correlation time as the number of snapshots that can be predicted while maintaining a certain pearson correlation $\tau$. This metric, demonstrates what methods are capable of performing stable autoregressive rollouts without drifting too far from the ground truth. We report

Table 2: Evaluation for 5D turbulence modelling and nonlinear heat flux prediction. We evaluate all methods across six in-distribution (**ID**) and five out-of-distribution (**OOD**) simulations. For $\bar{Q}$ we report RMSE of time-averaged predictions after an autoregressive rollout. For $f$ we report correlation time for autoregressive rollouts with threshold $\tau = 0.1$. Higher correlation time is better.

| Method | Input | $f$ | | $\bar{Q}$ | |
|---|---|---|---|---|---|
| | | **ID** (↑) | **OOD** (↑) | **ID** (↓) | **OOD** (↓) |
| *SOTA Reduced Numerical modelling* | | | | | |
| QL (Bourdelle et al., 2007) | 3D | n/a | n/a | $89.48 \pm 11.77$ | $95.18 \pm 21.58$ |
| *Classical Regression Techniques* | | | | | |
| GPR (Hornsby et al., 2024) | 0D | n/a | n/a | $43.82 \pm 10.84$ | $59.28 \pm 17.55$ |
| MLP | 0D | n/a | n/a | $50.90 \pm 10.87$ | $62.88 \pm 18.58$ |
| *Neural Surrogate Models (48 simulations)* | | | | | |
| FNO (Li et al., 2021) | 3D | $9.33 \pm 0.56$ | $9.20 \pm 0.58$ | $119.88 \pm 13.15$ | $124.96 \pm 23.27$ |
| PointNet (Qi et al., 2016) | 5D | $8.5 \pm 1.26$ | $9.0 \pm 0.89$ | $119.90 \pm 13.16$ | $125.01 \pm 23.28$ |
| Transolver (Wu et al., 2024) | 5D | $9.66 \pm 0.94$ | $10.2 \pm 0.75$ | $119.92 \pm 13.15$ | $125.02 \pm 23.28$ |
| ViT (Dosovitskiy et al., 2021) | 5D | $16.83 \pm 1.49$ | $19.20 \pm 1.36$ | $119.63 \pm 13.13$ | $125.13 \pm 23.29$ |
| GyroSwin (Ours) | 5D | $26.50 \pm 3.55$ | $28.60 \pm 8.82$ | $67.68 \pm 10.28$ | $70.48 \pm 17.21$ |
| *Scaling GyroSwin to 241 simulations* | | | | | |
| GyroSwin$_{Small}$ (Ours) | 5D | $98.00 \pm 27.53$ | $76.40 \pm 17.60$ | $23.72 \pm 4.05$ | $53.54 \pm 18.10$ |
| GyroSwin$_{Medium}$ (Ours) | 5D | $94.17 \pm 21.96$ | $91.20 \pm 18.61$ | $37.24 \pm 9.60$ | $44.17 \pm 17.68$ |
| GyroSwin$_{Large}$ (Ours) | 5D | $\mathbf{110.33 \pm 19.74}$ | $\mathbf{111.80 \pm 23.86}$ | $\mathbf{18.35 \pm 1.56}$ | $\mathbf{26.43 \pm 9.49}$ |

correlation times for the neural surrogates in Table 2. We observe a clear trend that GyroSwin is by far the most stable autoregressive method compared to other neural surrogates. Remarkably, when scaling GyroSwin in terms of data and model size, we attain stable rollouts for over 100 timesteps, even for **OOD** simulations.

**5D→0D average flux.** The task for this evaluation is to predict the average heat flux $\bar{Q}$ over the last 80 timesteps of a simulation from the full 5D field after an autoregressive rollout. We present results for both ID and OOD evaluation sets in Table 2. On the reduced training set GyroSwin yields the best performance compared to the QL model and alternative 5D neural surrogates. Interestingly, most neural surrogates converge to similar heat flux $\bar{Q}$ of around 120. This value is obtained if the model diverges and the flux predictions for each simulation are consistently zero, which would be the case if no turbulence occurs at all. This indicates that all competitors fail to capture turbulent transport as they do not predict any turbulent flux transport after longer rollouts, leading to similar errors for $\bar{Q}$. Furthermore, when scaling GyroSwin to more data and larger model sizes, we observe a drastic improvement in nonlinear flux prediction, achieving significantly lower error than currently existing surrogates. In Appendix G we also show that the composition of all components of GyroSwin results in the best performance.

**Scalability.** In integrated plasma simulations, turbulent heat fluxes must be repeatedly computed across thousands of runs, radial locations, and time intervals, underscoring the need for scalable surrogate architectures. To assess scalability of neural surrogates, we report inference speed, memory consumption, and number of parameters on a single NVIDIA H100 80GB HBM3 in Table 3. We exclude the 3D FNO as it is based on collapsing velocity dimensions into channels (see Appendix D) which is inherently unscalable to higher resolutions. Factorized variants of FNO and CNN are also slow and memory heavy. Field-based surrogates (PointNet, Transolver) require subsampling during training and chunked inference, limiting scalability. ViTs suffer from quadratic complexity and similar memory consumption as GyroSwin despite using half of its parameters. Scaling ViT to the same parameter count as GyroSwin$_{Small}$ results in $\sim$ 18ms inference speed which is $52.5\%$ slower than GyroSwin$_{Small}$. GyroSwin$_{Small}$ is roughly three orders of magnitude faster than the numerical code GKW (4200 vs. 756 GFLOPs). When scaled to $\sim$1B parameters (Figure 3), GyroSwin continues to improve training and validation loss for both $f$ and $\phi$ prediction, demonstrating superior scalability and strong potential for scaling to higher-fidelity simulations.

**5D→1D flux spectrum.** A major advantage of quasilinear approaches over tabular regressors is physical verifiability. As they are based on 3D linear simulations, flux contributions for each mode

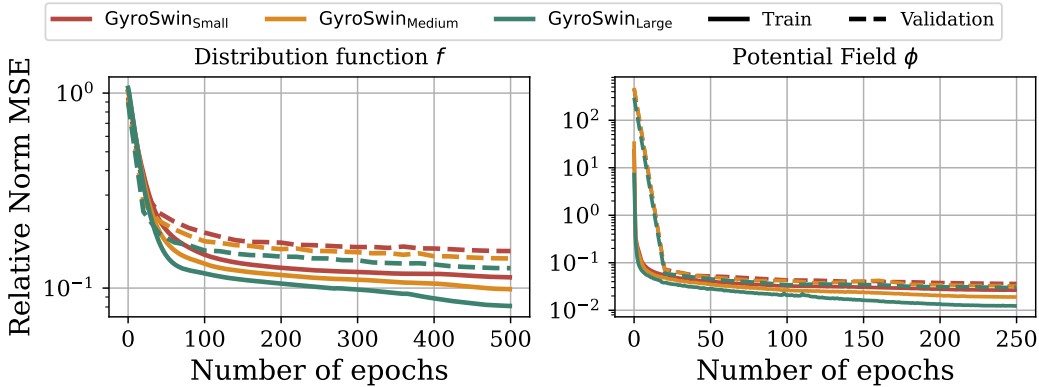

Figure 3: Scaling GyroSwin to ∼1B parameters trained on 241 simulations amounting to approximately 6TB of data. We show train/validation error for predicting the 5D distribution function (left) and the 3D electrostatic potential field (right).

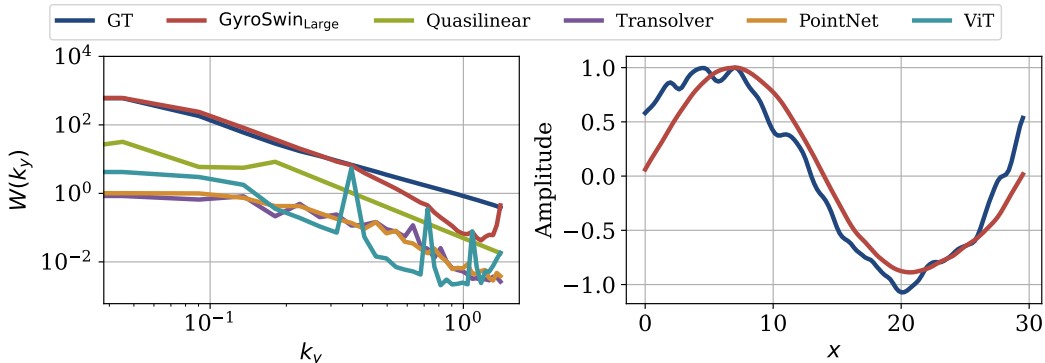

Figure 4: **Left:** $W(k_y)$ averaged over time and **OOD** simulations for different 5D neural surrogates. Competitors tend to underestimate while GyroSwin matches the spectrum well with a slight discrepancy on higher frequencies. **Right:** Time-averaged zonal flow profile for a slice along $s$ across radial coordinates $x$ for a selected **OOD** simulation. GyroSwin captures the zonal flow profile.

in $k_y$ ($Q(k_y)$) from Equation (3) can be inspected. This provides insights as to whether modes are captured that contribute most to heat transport. Similarly, we assess whether the general structure of $Q(k_y)$ is captured in predicted 5D snapshots of neural surrogates. We report pearson correlation to the ground-truth $Q(k_y)$ for all 5D neural surrogates in Table 3. We observe that all neural surrogates exhibit higher correlation than the QL approach highlighting their potential. Furthermore, we demonstrate that GyroSwin exhibits the best correlation on the **OOD** test cases. However, it performs slightly worse on the **ID** cases than competitors. When scaling GyroSwin to the large training set, we find a boost in performance, resulting in almost perfect reproduction of the shape of $Q(k_y)$. This means that GyroSwin$_{\text{Large}}$ captures the energy transport per-mode almost perfectly. In Figure 10 we visualize $Q(k_y)$ for all **ID** as well as **OOD** cases.

As mentioned in Section 2 another useful diagnostic is the turbulence intensity spectrum $W(k_y)$ which is not necessarily aligned with $Q(k_y)$, i.e. the mode transporting the most energy might not be the same where turbulence is most intense. Therefore we visually showcase $W(k_y)$ in Figure 4 (left) averaged across time and all **OOD** cases. In addition we provide visualizations for each test case of **ID** and **OOD** separately in Figure 9. We observe that while all neural surrogates decently reproduce the shape of the spectrum, they heavily underestimate the magnitude. Furthermore, GyroSwin$_{\text{Large}}$ matches shape and magnitude almost perfectly, up until a few higher frequency components. We attribute this finding to the spectral bias of neural networks towards lower frequency (higher energy) modes (Rahaman et al., 2019).

Table 3: Evaluation for scalability, diagnostics, and nonlinear physics. We evaluate all methods across six in-distribution (**ID**), and five out-of-distribution (**OOD**) simulations. For diagnostics and nonlinear physics we report time-averaged pearson correlation for autoregressive rollout. GyroSwin is the only method that can be scaled to ~1B parameters while maintaining reasonable inference speed and improving correlation to diagnostics.

| Method | Fwd[ms] | Mem[GB] | Params[M] | $Q(k_y)$ ID ($\uparrow$) | OOD ($\uparrow$) |
|---|---|---|---|---|---|
| *SOTA Reduced Numerical modelling* | | | | | |
| QL (Bourdelle et al., 2007) | N/A | N/A | N/A | $0.51 \pm 0.4$ | $0.58 \pm 0.33$ |
| *Neural Surrogate Models (48 simulations)* | | | | | |
| F-CNN | 569.6 | 11.1 | 2.0 | N/A | N/A |
| F-FNO | 963.3 | 36.9 | 1.3 | N/A | N/A |
| PointNet (Qi et al., 2016) | 19.7 | 39.7 | 43.9 | $0.49 \pm 0.02$ | $0.54 \pm 0.09$ |
| Transolver (Wu et al., 2024) | 3103 | 35.8 | 40.6 | $0.50 \pm 0.02$ | $0.53 \pm 0.09$ |
| ViT (Dosovitskiy et al., 2021) | 6.3 | 2.4 | 46.1 | $0.52 \pm 0.05$ | $0.63 \pm 0.11$ |
| GyroSwin (Ours) | 11.8 | 2.8 | 90.2 | $0.59 \pm 0.06$ | $0.69 \pm 0.06$ |
| *Scaling GyroSwin to 241 simulations* | | | | | |
| GyroSwin$_{Small}$ (Ours) | 11.8 | 2.8 | 90.2 | $0.82 \pm 0.10$ | $0.79 \pm 0.09$ |
| GyroSwin$_{Medium}$ (Ours) | 12.1 | 5.3 | 250.9 | $0.72 \pm 0.08$ | $0.76 \pm 0.12$ |
| GyroSwin$_{Large}$ (Ours) | 15.4 | 9.6 | 998.3 | $\mathbf{0.87} \pm 0.11$ | $\mathbf{0.91} \pm 0.08$ |

**New capability: 5D zonal flow modelling.** Zonal flows have a significant impact on the turbulence dynamics. To evaluate whether GyroSwin is capable of capturing zonal flows, we visualize the zonal flow profile of a test case of the **OOD** set. We again time-average the profiles and compare to the time-averaged predicted zonal flow profile of GyroSwin$_{Large}$. We find that GyroSwin$_{Large}$ accurately captures the zonal flow profile. This capability has so far been unreachable by any other surrogate modelling technique. We provide additional visualizations for each test case of the **ID** and **OOD** sets in Figure 11.

## 6 Conclusions and Limitations

We present GyroSwin, a scalable neural surrogate model for nonlinear gyrokinetic equations modelling turbulent transport in Plasmas. Unlike existing surrogate models, GyroSwin operates directly in a 5D space and evolves the 5D distribution function of gyrokinetics. It is based on latent cross-attention and integration modules for 3D↔5D latent interaction and trained in a multitask fashion to predict the distribution function, electrostatic potentials and heat fluxes. Furthermore, we perform channelwise mode separation to incorporate an inductive bias towards essential nonlinear phenomena (zonal flows) that are essential for modelling turbulence. We show that GyroSwin outperforms reduced numerical appraoches and scales favorably compared to other neural surrogates. Furthermore, GyroSwin accurately captures nonlinear phenomena self-consistently, a capability not present in prior surrogate or reduced numerical models. As a result, GyroSwin offers a fruitful alternative to efficient approximation of turbulent transport.

Currently, the main limitation of GyroSwin is that it does not take into account the chaotic and therefore distributional nature of turbulence. Although we find that GyroSwin produces stable autoregressive rollouts over 100 timesteps, it suffers from error accumulation. Generative modelling offers a remedy to this problem by directly predicting snapshots in the saturated phase. We aim to incorporate such distributional approaches in future work. Furthermore, we neglect the linear phase of the simulation. The reason for this is that our focus primarily lies in modelling turbulence. In the future we aim to extend GyroSwin to the linear phase as well. Finally, we only consider the adiabatic electron approximation, as it allows generation of a relatively large training set at rather low cost. Still, each simulation produces considerable data volumes, making full coverage of the 4D parameter space difficult. Beyond that, we envision a high-fidelity surrogate model by transfer learning from low-fidelity approximate simulations.

## Acknowledgments and Disclosure of Funding

We extend special thanks to our colleagues Wei Lin for the discussion on computer vision and multi-task learning, and Anna Zimmel and Gerald Gutenbrunner for conversations on the spectral behavior of neural networks.

This work has been funded by the Fusion Futures Programme. As announced by the UK Government in October 2023, Fusion Futures aims to provide holistic support for the development of the fusion sector. The ELLIS Unit Linz, the LIT AI Lab, the Institute for Machine Learning, are supported by the Federal State Upper Austria. We thank the projects FWF AIRI FG 9-N (10.55776/FG9), AI4GreenHeatingGrids (FFG- 899943), Stars4Waters (HORIZON-CL6-2021-CLIMATE-01-01), FWF Bilateral Artificial Intelligence (10.55776/COE12). We acknowledge EuroHPC Joint Undertaking for awarding us access to Leonardo at CINECA, Italy, and Deucalion at MACC, Portugal.

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

# A  Derivation of the Gyrokinetic equation

We begin with the Vlasov equation for the distribution function $f(\boldsymbol{r}, \boldsymbol{v}, t)$:

$$\frac{\partial f}{\partial t} + \mathbf{v} \cdot \nabla f + \frac{q}{m} \left( \boldsymbol{E} + \boldsymbol{v} \times \boldsymbol{B} \right) \cdot \nabla_v f = 0 \tag{9}$$

The Vlasov equation describes the conservation of particles in phase space in the absence of collisions. Here, $\boldsymbol{r} = (x, y, s)$ and $\boldsymbol{v} = (v_x, v_y, v_s)$ correspond to coordinates in the spatial and the velocity domain, respectively. Hence the Vlasov equation is a 7D (including time) PDE representing the density of particles in phase space at position $\boldsymbol{r}$, velocity $\boldsymbol{v}$, and time. The term $\nabla_v f$ describes the response of the distribution function to accelerations of particles and $\frac{q}{m} \left( \mathbf{E} + \mathbf{v} \times \mathbf{B} \right)$ denotes the Lorentz force, which depends on particle charge $q$ and mass $m$, as well as electric field $\boldsymbol{E}$ and magnetic field $\boldsymbol{B}$. Finally, the advection (or convection) term $\boldsymbol{v}\nabla f$ describes transport of the distribution functon through space due to velocities.

To derive the *gyrokinetic equation*, we transform from particle coordinates to guiding center coordinates $(\mathbf{R}, v_\parallel, \mu, \theta)$, where $\mu = \frac{mv_\perp^2}{2B}$ is the magnetic moment, $\theta$ the gyrophase, which describes the position of a particle around its guiding center as it gyrates along a field line, and $\boldsymbol{R}$ is the coordinate of the guiding center.

Assuming the time scale $L$ at which the background field changes is much longer than the gyroperiod with a small Larmor radius $\rho \ll L$, we can *gyroaverage* to remove the dependency on the gyrophase $\theta$, yielding:

$$\frac{\partial f}{\partial t} + \dot{\boldsymbol{R}} \cdot \nabla f + \dot{v}_\parallel \frac{\partial f}{\partial v_\parallel} = 0 \tag{10}$$

**Linear Terms**

The unperturbed (background) motion of the guiding center is governed by:

$$\dot{\boldsymbol{R}} = v_\parallel \boldsymbol{b} + \boldsymbol{v}_D \tag{11}$$

$$\dot{v}_\parallel = -\frac{\mu}{m} \boldsymbol{b} \cdot \nabla \boldsymbol{B} \tag{12}$$

Here, $\mathbf{b} = \boldsymbol{B}/B$ is the unit vector along the magnetic field, and $\boldsymbol{v}_D$ represents magnetic drifts. Substituting into the kinetic equation yields

$$\frac{\partial f}{\partial t} + (v_\parallel \boldsymbol{b} + \boldsymbol{v}_D) \cdot \nabla f - \frac{\mu}{m} \boldsymbol{b} \cdot \nabla \boldsymbol{B} \frac{\partial f}{\partial v_\parallel} = 0 \tag{13}$$

We can express the magnetic gradient term using:

$$\boldsymbol{b} \cdot \nabla \boldsymbol{B} = \frac{\boldsymbol{B} \cdot \nabla \boldsymbol{B}}{B} \tag{14}$$

so that:

$$\frac{\mu}{m} \boldsymbol{b} \cdot \nabla \boldsymbol{B} = \frac{\mu B}{m} \frac{\boldsymbol{B} \cdot \nabla \boldsymbol{B}}{\boldsymbol{B}^2} \tag{15}$$

**Nonlinear Term**

Fluctuating electromagnetic potentials $\delta\phi, \delta\boldsymbol{A}$ induce E×B and magnetic flutter drifts. We define the gyroaveraged generalized potential as

$$\chi = \langle \phi - \frac{v_\parallel}{c} \boldsymbol{A}_\parallel \rangle, \tag{16}$$

where $\boldsymbol{A}_\parallel$ is the aprallel component of the vector potential, $\langle \cdot \rangle$ denotes the gyroaverage, and $c$ is the speed of light, which is added to ensure correct units. $\phi$ is the electrostatic potential, the computation of which involves an integral of $f$ over the velocity space (see eq. 1.41 in the GKW manual [3] for a complete description).

---

[3] https://bitbucket.org/gkw/gkw/src/develop/doc/manual/

This gives rise to the drift

$$\mathbf{v}_\chi = \frac{c}{B}\mathbf{b} \times \nabla\chi, \tag{17}$$

and yields the nonlinear advection term $\mathbf{v}_\chi \cdot \nabla f$.

**Final Equation**

We arrive at the gyrokinetic equation in split form:

$$\underbrace{\frac{\partial f}{\partial t} + (v_\parallel \mathbf{b} + \mathbf{v}_D) \cdot \nabla f - \frac{\mu B}{m}\frac{\mathbf{B} \cdot \nabla B}{B^2}\frac{\partial f}{\partial v_\parallel}}_{\text{Linear}} + \underbrace{\mathbf{v}_\chi \cdot \nabla f}_{\text{Nonlinear}} = S \tag{18}$$

Here, $S$ represents external sources, collisions, or other drive terms. To enhance the tractability of Equation (1), the distribution function $f$ is usually split into equilibrium and perturbation terms

$$f = f_0 + \delta f = \underbrace{f_0 - \frac{Z\phi}{T}f_0}_{\text{Adiabatic}} + \underbrace{\frac{\partial h}{\partial t}}_{\text{Kinetic}}, \tag{19}$$

where $f_0$ is a background or equilibrium distribution function, $T$ the particle temperature, $Z$ the particle charge, $\phi$ the electrostatic potential, and $\delta f$ the total perturbation to the distribution function, which comprises of *adiabatic* and *kinetic* response. The adiabatic term describes rapid and passive responses to the electrostatic potential that do not contribute to turbulent transport, while the kinetic term governs irreversible dynamics that facilitate turbulence. Numerical codes, such as GKW (Peeters et al., 2009), rely on solving for $\delta f$ instead of $f$. A common simplification is to assume that electrons are adiabatic, which allows us to neglect the kinetic term in the respective $\delta f$. Hence, the respective $f$ for electrons ($f_e$) does not need to be modelled, effectively halving the computational cost.

## B   Quasilinear models

We used the QuaLiKiZ saturation rule (Bourdelle et al., 2007) applied to lienar GKW simulations, following Kumar et al. (2021). The QuaLiKiz saturation rule estimates turbulent transport based on linear gyrokinetic stability analysis, using quasilinear theory and empirical saturation rules. The quasilinear ion heat flux $Q$ is modelled as:

$$Q = \sum A_k W_k \tag{20}$$

where $A_k$ is the *linear weight spectrum* that quantifies the phase relationship between electrostatic potential fluctuations and that is retrieved by solving the linear gyrokinetic equation, and $W_k$ is the *turbulence intensity spectrum* which in QuaLiKiz is parametrised by a shape function $S_k$ and a normalisation factor $C$ calibrated on nonlinear gyrokinetic simulations.

To fit the normalisation factor $C$, we assume access to a vector of nonlinear fluxes $\boldsymbol{q} \in \mathbb{R}^j$ and set the right-hand side of Equation (20) to $\boldsymbol{x} \in \mathbb{R}^j$, then the optimal solution for $C$ can be obtained via a least-squares fit

$$C = \frac{\sum_j x_j q_j}{\sum_j x_i^2}. \tag{21}$$

In our setup, we used flux estimates from the small training set (48 simulations) to compute the fit that resulted in a $C = 7.93$. This value differs slightly from the one reported in Kumar et al. (2021). We attribute the difference to the different sampling strategy of simulations. The QuaLiKiz saturation rule retains key physics from gyrokinetics while allowing efficient prediction of turbulent transport in integrated modelling frameworks. Furthermore, we found it beneficial to center the nonlinear flux vector $\boldsymbol{q}$ which alters the normalisation constant $C$, but leads to reduced error on flux predictions on a separate validation set.

## C   Data Generation and Visualisation

**Data Generation.**

In this work, we mainly consider turbulence driven by ion temperature gradients, also typically referred to as the cyclone base case (Dimits et al., 2000a). We leverage the numerical code GKW (Peeters et al., 2009) for generating nonlinear and linear gyrokinetic simulations, by varying four parameters: $R/L_t$, $R/L_n$, $\hat{s}$, and $q$, which significantly affect emerging turbulence in the Plasma. $R/L_t$ is the ion temperature gradient, which is the main driver of turbulence. $R/L_n$ is the density gradient, which has a more stabilizing effect but can sometimes also lead to enhanced turbulence. The parameter $q$ denotes the so-called safety factor, which is the inverse of the rotational transform and describes how often a particle takes a poloidal turn before taking a toroidal turn. $\hat{s}$ denotes magnetic shearing. It has a stabilizing effect as more magnetic shearing results in improved isolation of the Plasma.

In total we perform two data generation passes. For the first pass, we specify the ranges to sample the four parameters as $R/L_T \in [3, 12]$, $R/L_n \in [1, 7]$, $q \in [1, 9]$, and $\hat{s} \in [0.5, 5]$. In addition, we also always vary the noise amplitude of the initial condition within $[1e - 5, 1e - 3]$ and the initial condition itself as one of $[\sin, \cos, \text{random}]$. We generate 100 simulations in total with this setup, but found that only approximately half of them actually exhibit turbulence. The turbulent configurations are depicted as the sparse point cloud in Figure 5. Therefore, we adapt the parameter ranges for the second data generation pass to $R/L_T \in [6, 12]$, $R/L_n \in [0, 2]$, $q \in [5, 9]$, and $\hat{s} \in [0.5, 2]$. Effectively, this change in parameters reduces stabilizing factors such as magnetic shearing and density gradients to force simulations to exhibit turbulence. We sample another 200 simulations from these parameter ranges and found that all of these develop strong turbulence (see distribution of $\bar{Q}$ in Figure 5). Eventually, we end up with around 250 simulations of which we randomly select three as a validation set during training and six as in-distribution evaluation set, to end up with a final training set of 241 simulations. Importantly, we still include the initially generated turbulent simulations as data generation is computationally expensive. Finally, we use the same parameter combination of each nonlinear simulation to generate their linear counterparts for the reduced-order model.

**5D Data Visualization.** We show a visual illustration of the 5D distribution function in Figure 6. For visualization purposes, we always show combinations of the different axes while averaging or slicing across the remaining ones. This way, we end up with 2D planes that are easy to visualize.

Furthermore, we also highlight the difference between nonlinear and linear simulations in Figure 7. We plot 2D planes of the 5D field of nonlinear and linear simulations side-by-side for three different timesteps. Since colorbars are shared across each pair of 2D planes, the difference between nonlinear and linear simulations can be clearly observed. Specifically, the structure in the linear simulation is symmetric, while the nonlinear simulation exhibits asymmetries. Furthermore, the difference is particularly obvious in wavenumber space, where (1) magnitudes of nonlinear simulations are much larger, and (2), there is a lot more interaction between modes. Hence, there is structure in the nonlinear simulations that is not captured in linear ones, which is a drawback of approximations based on linear simulations, i.e. for quasilinear models.

## D   Implementation details

**Data preprocessing.** A first measure we take is standardizing all fields and fluxes to zero mean and unit variance. We observed that this positively affects training. To enable autoregressive rollouts, we simply accumulate all statistics across all training simulations, and use them to normalize and denormalize during inference. Furthermore, another measure that had significant impact on training dynamics is transferring from spectral $k_x, k_y$ coordinates to real $x, y$. This positively affects normalization as the difference in magnitudes is less pronounced in real space. While electrostatic potential fields coming from GKW are already in real space, they are padded for the real FFT and result in different spatial dimensions to the 5D distribution function. This is inconvenient for up/downsampling, so we unpad the Fourier-space electrostatic potentials to match spatial dimensions of the 5D field, before transforming them back to real space.

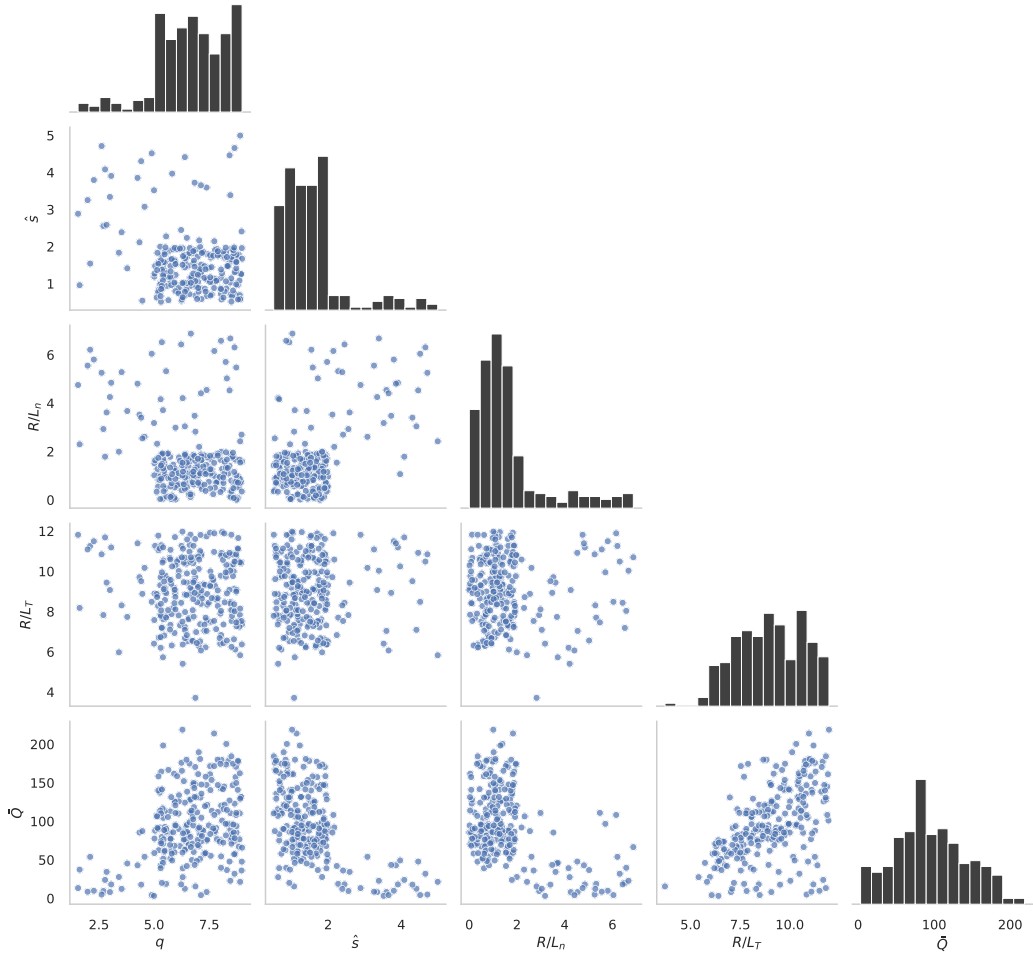

Figure 5: Distribution of input parameters $\hat{s}$, $q$, $R/L_n$, and $R/L_t$ along with average heat flux $\bar{Q}$. The sampled parameter space is evenly distributed.

**GyroSwin.** Vision Transformers usually employ convolutions for patch operations (Dosovitskiy et al., 2021; Liu et al., 2021, 2022), but they are not scalable to 5D. Whereas in convolutions the kernel is shared across every location, this is harder to parallelize in a general nD setting. Therefore, we employ less parameter-efficient *local* convolutions with independent kernels at each location, implemented through fully connected layers. The n-dimensional input grids are first tiled with adaptive reshapes, then the flattened patch dimension is embedded with a shallow MLP. Patch embedding, merging, and expansions are implemented as linear layers or MLPs. Furthermore, we add relative positional biases and condition all Swin layers on the 4 parameters as well as the current timestep via FiLM (Perez et al., 2018). We experimented with DiT-style conditioning (Peebles & Xie, 2023), but found no improvements over FiLM.

We train GyroSwin for next-step prediction of the 5D distribution function of nonlinear gyrokinetic simulations. We use the Adam optimizer (Kingma & Ba, 2015) with a weight decay of 1e-5 and a cosine learning rate scheduler with linear warmup and a peak at 3e-4, decayed to 0. During training we employ automatic mixed precision and cast to bfloat16 with gradient clipping to a magnitude of 1. Due to the bulk of training data we perform lazy dataloading which results in substantial overhead, however it is not possible to fit all data in RAM. In our special case of adiabatic electrons, each direction in the magnetic moment $\mu$ is independent of each other. Therefore, this dimension

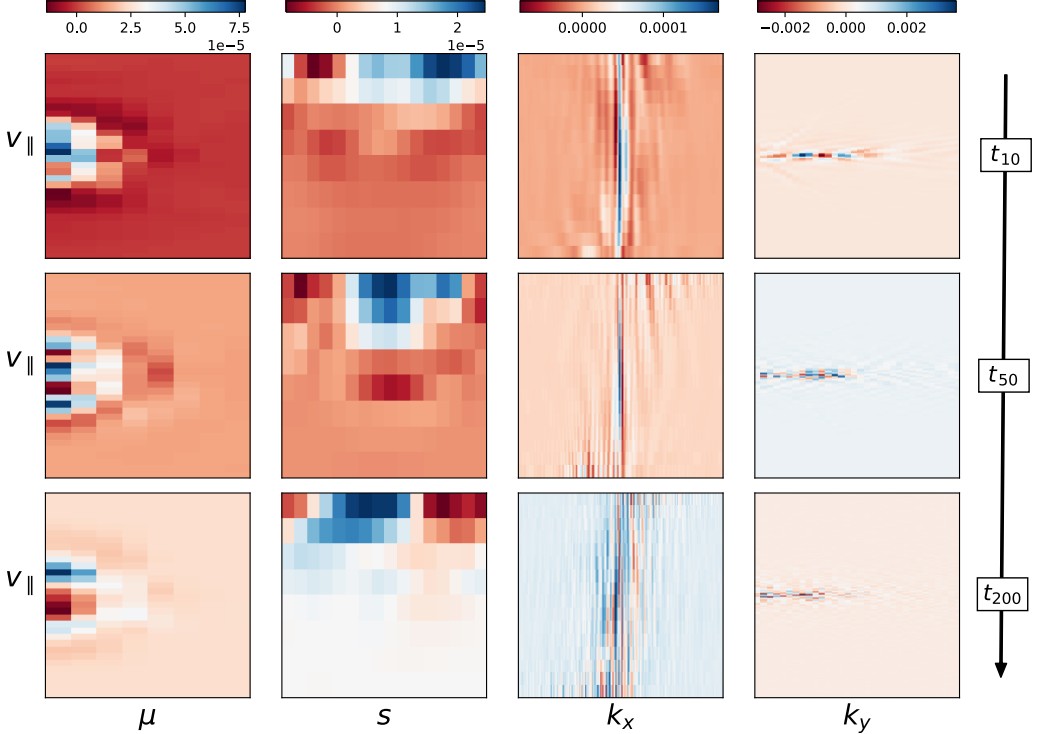

Figure 6: Visualization of the 5D distribution function of nonlinear gyrokinetics (ground truth, Fourier space along $k_x$ and $k_y$). We show different combinations of axes of the 5D field while averaging over the remaining ones for different timesteps. Colorbars are shared columnwise.

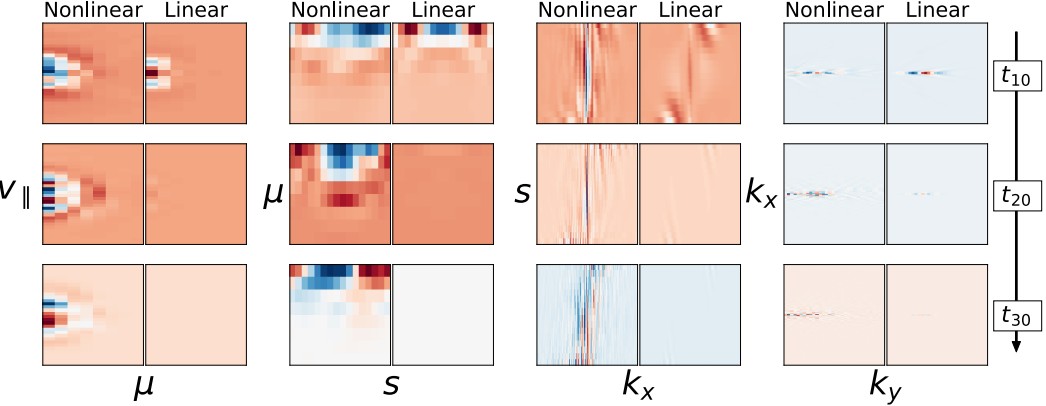

Figure 7: Comparison of 5D distribution functions for linear and nonlinear simulations (ground truth, Fourier space along $k_x$ and $k_y$). We show different combinations of axes of the 5D field while averaging over the remaining ones for different timesteps. Colorbars are shared across each 2D nonlinear and linear plane.

can be decoupled from the remaining ones and viewed as additional channels. This results in a substantial speedup without any loss of information.

For the smaller training set (48 simulations), we train our model for 200 epochs and evaluate every 20 epochs on the three holdout trajectories based on which we perform model selection. During inference, we roll out the model autoregressively for the entire duration of each simulation (185 timesteps). GyroSwin is trained on four H100 GPUs with 80GB VRAM using PyTorch's Distributed Data Parallel (DDP) for approximately 120 hours.

For the larger training set (241 simulations), we train GyroSwin for 500 epochs on 16 H100 GPUs with 80GB VRAM. We employ a scheduler for the different loss terms mentioned in Equation (8). Specifically, we start training only on $\mathcal{L}_f$ for the first half of training, then switch on $\mathcal{L}_\phi$ and linearly increase $w_\phi$ from 0 to 1 from epoch 250 to 350 and keep it to 1 afterwards. From epoch 350 onwards we also activate $w_{\bar{Q}}$ and again increase it linearly from 0 to 1 between epoch 350 and 450 and continue with $w_f = w_\phi = w_{\bar{Q}} = 1$ for the last 50 epochs. Depending on the model size, we trained up until 336 hours (GyroSwin$_\text{Large}$).

## D.1 Baselines

**Gaussian Process Regression (GPR).** We use a four-parameter GPR trained on the operating parameters ($R/L_T$, $R/L_n$, $q$, and $\hat{s}$). It uses a Matern $3/2$ kernel (Rasmussen & Williams, 2006) to predict the nonlinear flux. This is the model proposed by Hornsby et al. (2024).

**Multi-Layer Perceptron (MLP).** We use a 3 layer MLP with 128 hidden dimension and GELU activations. The inputs are the four operating parameters ($R/L_T$, $R/L_n$, $q$, and $\hat{s}$), which are embedded in a continuous sin-cos space, and the output is the time-averaged scalar heat flux. This is similar to QLKNN proposed by van de Plassche et al. (2020), but trained on nonlinear fluxes instead of quasiliner ones.

**FNO.** The `neuraloperator` library[4] (Kovachki et al., 2023) contains tensorized nD implementations of Fourier convolution layers. However, as shown in Table 3 these are too expensive to be trained on our data. To enable training FNO (Li et al., 2020), we utilize a much faster 3D FNO which operates only on the spatial dimensions, while flattening the velocity space in the channels. This baseline is shown in Table 2 and has 256 latent dimension, 4 layers and considers $\frac{1}{2}$ of the total modes on each dimension (5,004 flat spatial modes).

**PointNet.** We use PointNet's (Qi et al., 2016) implementation from SIMSHIFT (Setinek et al., 2025), adapted to work on regular grid data. The total number of points on the grid is approximately 22.3 million considering both channel dimensions. Therefore, to train this coordinate-based baseline, we randomly subsample the grid to 65,536 points for each sample. We also experimented with larger number of coordinates per-batch but observed diminishing returns beyond 65,536 points. Since the task is to predict the evolution of the distribution function, we provide its values at the corresponding coordinates at time $t$ to predict those for $t + 1$. For embedding grid coordinates we employ per-dimension embedding layers of the corresponding resolution and concatenate them to obtain the final coordinate embedding. Each of these embeddings is of dimension 64 leading to a hidden dimension of 320 for 5D input. We use GELU activations and condition the concatenated values and coordinates on the four operating parameters. The encoder projects the conditioned coordinates/values to dimensionality 640. Finally, we project to 12,800 global pooling features which are concatenated with the local coordinate features and serve as input to the decoder that predicts the field values at the local coordinates for the next timestep.

**Transolver.** We use Transover's (Wu et al., 2024) implementation from SIMSHIFT (Setinek et al., 2025), adapted to work on regular grid data. We follow the same sampling and coordinate strategy as for PointNet. The Transolver embedding dimension is set to 512 with 8 transformer blocks using 8 attention heads and 256 slices per head. Again, we provide the values of the distribution function at time $t$ as input to predict the values at $t + 1$.

---

[4]https://github.com/neuraloperator/neuraloperator

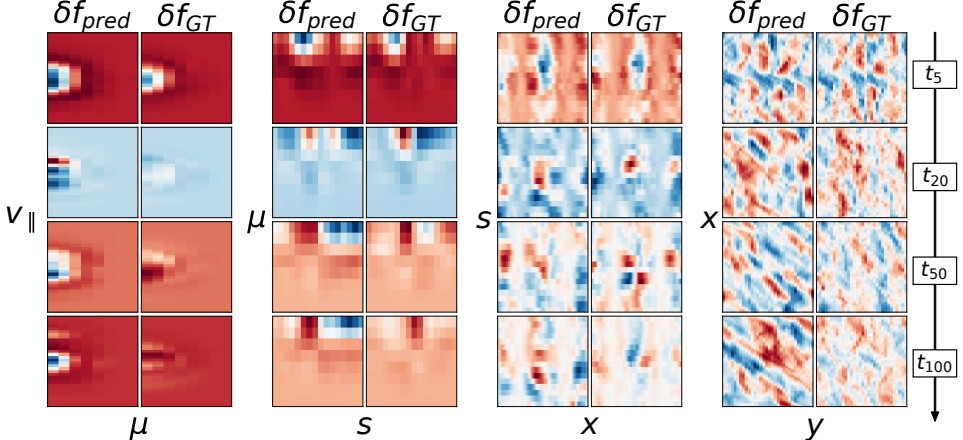

Figure 8: Side-by-side comparison of autoregressive rollout predictions with GyroSwin compared to ground truth ($x$ and $y$ in real space), in the saturated phase with shared colorbars. GyroSwin preserves the high-level structure within the first rollout steps. After a larger amount of rollout steps error accumulates, but predictions remain stable and do not diverge.

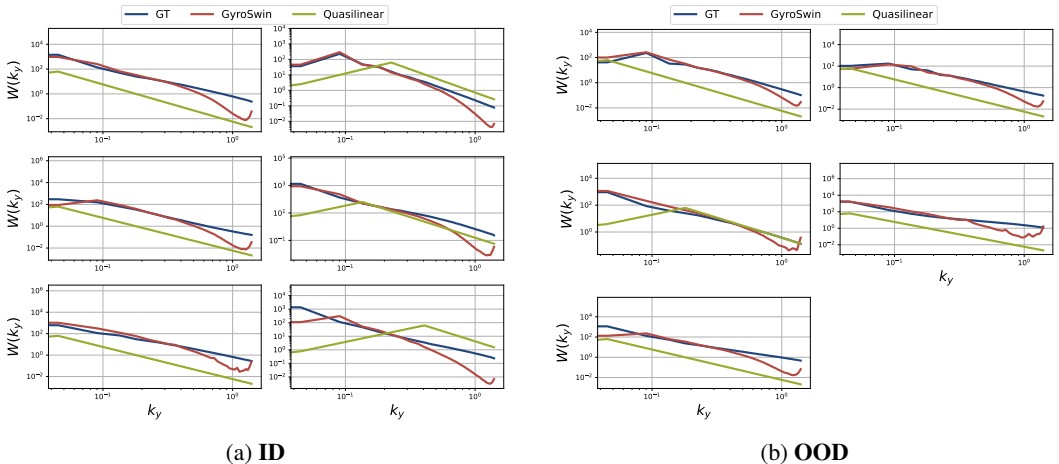

(a) **ID**                    (b) **OOD**

Figure 9: Comparison of $W(k_y)$ In-Distribution and Out-Of-Distribution.

## E    Additional Results

Finally, we include qualitative results for autoregressive rollouts of GyroSwin. Specifically, we visualize model predictions for $f$ for the test case $ID_1$ and compare it to the ground truth in Figure 8 for timesteps $\{10, 100, 200\}$. We observe that the overall structure is more or less preserved within the first 10 rollout steps, however, afterwards the model suffers from error accumulation. This is especially pronounced for the velocity space, where the model seems to overpredict. Interestingly, though, the predictions in wavenumber space remain somewhat stable and do not diverge, evan after 100 autoregressive prediction steps. Generally we can say that GyroSwin yields stable predictions for long rollouts well beyond 100 timesteps.

## F    Diagnostics

We provide visualizations for the time-averaged turbulence-intensity spectrum $W(k_y)$ for the six different **ID** test cases in Figure 9a and the five different **OOD** test cases in Figure 9b. Furthermore, we show flux spectra $Q(k_y)$ for the six different **ID** test cases in Figure 10a and the five different **OOD** test cases in Figure 10b. Generally, we observe that GyroSwin susbtantially improves over the state-of-the-art reduced-numerical quasilinear model on $W(ky)$, which is reflected in the pearson

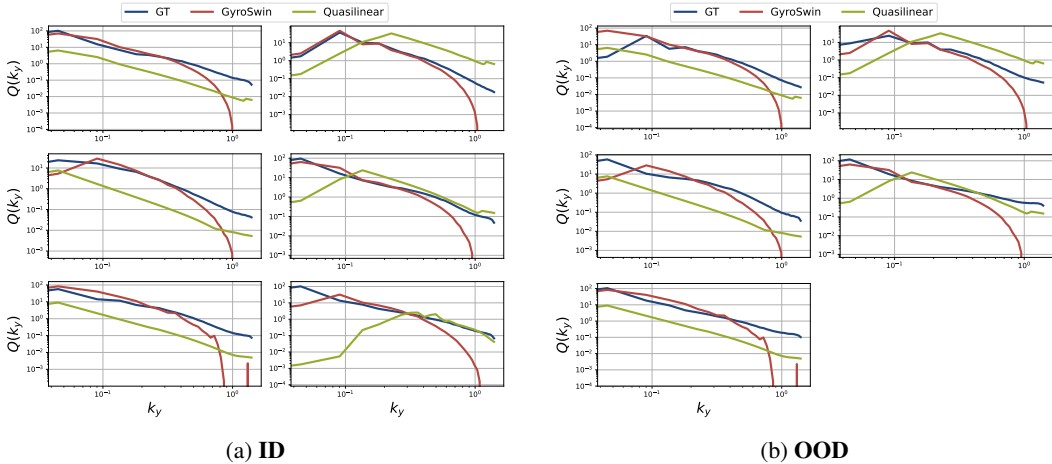

(a) **ID**  (b) **OOD**

Figure 10: Comparison of flux spectra In-Distribution and Out-Of-Distribution.

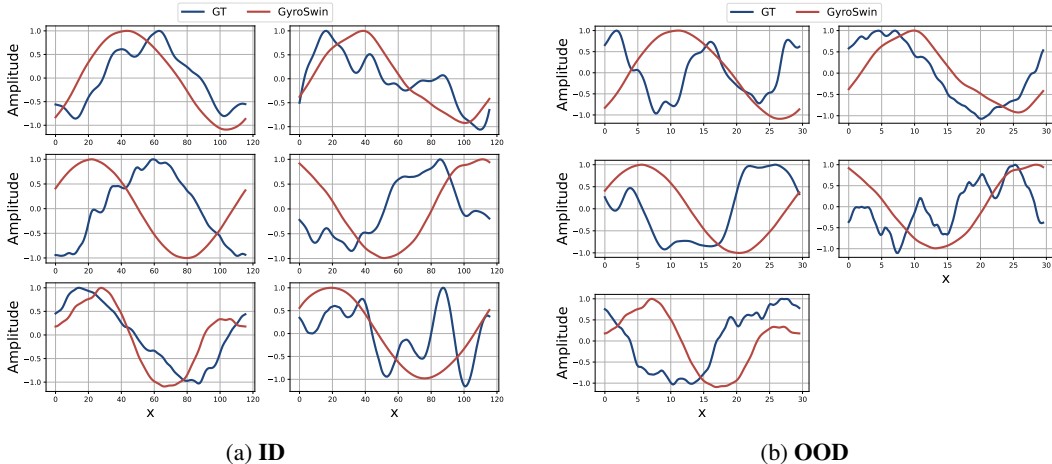

(a) **ID**  (b) **OOD**

Figure 11: Comparison of zonal flow profiles In-Distribution and Out-Of-Distribution.

correlation coefficients shown in Table 3. Interestingly, the fit for $Q(k_y)$ at first glance appears to be improved as well, however the correlation is approximately the same as for the quasilinear model. The reason for this is that high-frequencies are not well captured by GyroSwin. We expect this to be a result of the inherent spectral bias of neural network architectures (Rahaman et al., 2019).

In addition to the time-averaged spectra, we provide visualizations for the time-averaged zonal flow profile for all **ID** simulations as well as **OOD** simulations for GyroSwin$_{Large}$. We observe that GyroSwin$_{Large}$ usually tends to overestimate the amplitude therefore we normalize it for visualization purposes. The results for the **ID** and **OOD** test cases can be observed in Figure 11a, and Figure 11b, respectively. Generally, we observe that the predicted zonal flow profile of GyroSwin$_{Large}$ appears as a smoothed out version of the ground-truth. Again, our intuition is that this is due to the inherent low-frequency bias. Interestingly, on some test cases the zonal flow profile is entirely off, which explains the rather high variance observed in the main table. Overall, we can conclude that GyroSwin$_{Large}$ can capture the zonal flow of unseen simulations in an autoregressive manner even though there is plenty of room for improvement.

## G Ablation Studies

To justify the design choices of GyroSwin, we conduct a set of ablation studies as follows. We progressively add components of GyroSwin to a 5D Swin transformer and evaluate based on correlation

time to the 5D distribution function $f$ and RMSE for the time-averaged flux trace $\bar{Q}$. We report our results in Appendix G. Interestingly, the 5D-Swin already appears to be capable of performing stable rollouts, however it completely fails in capturing the nonlinear flux trace. When moving to a UNet like structure, we observe a slight drop in correlation time, but improved flux predictions. Furthermore, adding channelwise mode separation gives a boost in correlation time, particularly on **OOD** simulations. Moreover, there is a slight improvement in correlation time when adding our latent cross-attention and integrator modules. Finally, the most significant boost is observed when adding the flux prediction head, which results in our final design of GyroSwin, which achieves the best combination of stable autoregressive rollouts and nonlinear flux prediction.

Table 4: Ablation study on different components in GyroSwin, i.e. channel-wise mode separation of zonal flows and latent cross-attention/integrator modules. We report correlation time with $\tau = 0.1$ for the 5D distribution function and RMSE for $\bar{Q}$ on **ID** and **OOD** test cases. All methods are trained on a reduced dataset of 48 simulations.

| Method | Params [M] | $f$ | | $\bar{Q}$ | |
|---|---|---|---|---|---|
| | | **ID** ($\uparrow$) | **OOD** ($\uparrow$) | **ID** ($\downarrow$) | **OOD** ($\downarrow$) |
| 5D-Swin | 46.2 | $24.83 \pm 3.19$ | $25.00 \pm 2.39$ | $121.34 \pm 13.36$ | $125.74 \pm 23.23$ |
| GyroSwin$_f$ | 36.3 | $19.50 \pm 1.86$ | $18.60 \pm 0.51$ | $119.85 \pm 13.15$ | $124.91 \pm 23.2$ |
| GyroSwin$_f$ + ZF channel | 44.2 | $19.33 \pm 1.84$ | $22.00 \pm 2.26$ | $119.19 \pm 13.65$ | $122.76 \pm 23.63$ |
| GyroSwin$_{f+\phi}$ | 77.8 | $20.17 \pm 0.75$ | $21.80 \pm 1.20$ | $106.26 \pm 14.09$ | $112.95 \pm 25.22$ |
| GyroSwin$_{f+\phi}$ + LatentCA | 87.3 | $21.00 \pm 2.62$ | $22.80 \pm 2.89$ | $116.49 \pm 12.81$ | $121.41 \pm 23.45$ |
| GyroSwin$_{f+\phi+\bar{Q}}$ | 91.1 | $22.83 \pm 2.63$ | $25.00 \pm 0.71$ | $67.68 \pm 10.28$ | $70.48 \pm 17.21$ |

# H  Learning a saturation rule

We perform an additional ablation study where we discard the nonlinear term in Equation (1), e.g. train GyroSwin on linear simulations to predict nonlinear fluxes. This effectively mirrors a saturation in the Quasilinear setting, except that it does not rely on growth rates and reconstruction of the $k_y$ spectrum. Since linear simulations essentially converge to a fixed structure in the statistically steady state, we only use the last snapshot of a simulation and provide it as input to the model to predict the average nonlinear flux $\bar{Q}$. We call this variant GyroSwin$_{\text{Linear}}$ and compare it to the Quasilinear saturation rule as well as the tabular regressors. Since this baseline is relatively cheap to train, we train it for three seeds and for both training sets, containing 48 and 241 simulations and report our results in Equation (5).

Interestingly, we observe a consistent trend that machine learning based methods outperform the reduced numerical model. An interesting aspect of this result is that the quasilinear saturation rule leverages additonal data such as linear growth rates of modes and flux contributions per mode which are hidden to tabular regressors and the learned saturation rule. Furthermore, with increasing training data GyroSwin$_{\text{Linear}}$ outperforms tabular regressors GPR and MLP that map from operating parameters to nonlinear heat flux. Since GyroSwin$_{\text{Linear}}$ is only trained on the linear part of Equation (1), therefore we can conclude that it is capable of generalizing over linear mode structures to infer nonlinear fluxes. Finally, all methods experience improvements for larger training sets. For the QL model the improvement can be traced back to a larger scaling factor that is fitted on more simulations, whereas all other methods are machin-learning based, hence naturally improve with more available training data.

## H.1  Vision Transformer Complexity

Computational complexity of 5D swin layers, with an input of resolution $x \times y \times z \times h \times k$, $d$ channels and window size $M$ (assumed square for simplicity) is

$$O(\texttt{MSA}) = 4 \, (xyzhw)^2 \, d^2 + 2 \, (xyzhw)^2 \, d$$
$$O(\texttt{W-MSA}) = 4 \, xyzhw \, d^2 + 2M^5 \, xyzhw \, d$$

removing the squared dependency on the sequence length $xyzhw$, and replacing it with a much smaller window complexity of $M^5$. This consideration is the motivation behind the original swin paper by (Liu et al., 2021), as it makes attention on images close to linear in resolution.

Table 5: Learning a saturation rule with GyroSwin. We report RMSE for the average flux $\bar{Q}$ on **ID** and **OOD** test cases. GyroSwin$_{\text{Linear}}$ significantly outperforms the reduced numerical model (QL).

| Method | 48 sims | | 241 sims | |
|---|---|---|---|---|
| | **ID** ($\downarrow$) | **OOD** ($\downarrow$) | **ID** ($\downarrow$) | **OOD** ($\downarrow$) |
| *Reduced numerical methods* | | | | |
| QL | $89.48 \pm 11.77$ | $95.18 \pm 21.58$ | $56.68 \pm 14.09$ | $56.06 \pm 14.35$ |
| *Tabular regressors* | | | | |
| GPR | $43.82 \pm 10.84$ | $59.28 \pm 17.55$ | $36.95 \pm 9.24$ | $25.89 \pm 8.78$ |
| MLP | $50.90 \pm 10.87$ | $62.88 \pm 18.58$ | $49.54 \pm 13.36$ | $34.99 \pm 12.80$ |
| *Learned saturation rule* | | | | |
| GyroSwin$_{\text{Linear}}$ | $76.74 \pm 11.79$ | $82.95 \pm 19.58$ | $42.85 \pm 12.65$ | $23.40 \pm 5.89$ |

