# OpenReview forum: "GyroSwin: 5D Surrogates for Gyrokinetic Plasma Turbulence Simulations"
_NeurIPS.cc/2025/Conference — NeurIPS 2025 poster_

### Official Review · Reviewer_vuY7 · 2025-07-02

**Clarity:** 3
**Significance:** 2
**Originality:** 2
**Rating:** 4
**Confidence:** 3

**Summary:**

This paper presents GyroSwin, a deep learning model for simulation of 5D gyrokinetic plasma turbulence, a challenging problem crucial for fusion reactor design but is computationally prohibitive for traditional direct numerical simulation methods. Current reduced models, like Quasilinear (QL) approaches, are faster but neglect important nonlinear phenomena. GyroSwin tries to instead directly learn the evolution of the 5D distribution function to improve prediction accuracy. Modeling 5D fields is challenging using existing deep learning approaches such as CNNs or Vision Transformers (ViTs), so the modeling is based on Swin Transformers extended to 5D. It also uses physics-informed features such as integration of latent variables (reminiscent of Latent ODEs and traditional reduced order models). The model is trained in a multitask setup to predict the future 5D distribution, 3D potential fields and scalar heat flux. Experiments compare various in-distribution and out-of-distribution datasets and show that GyroSwin is comparable or the most accurate compared to standard DL, and non DL approaches.

**Questions:**

1. OOD Qi Performance: Why could explain why GyroSwin struggles in some OOD cases compared to GPR/MLP (Table 2), even though the modeling of GyroSwin has significantly more consideration for the underlying physics? I would have expected GyroSwin would overfit less.

1. While Fig 3 shows a successful rollout, would it be possible to visualize the distribution of errors over multiple rollouts as a function of time (different dataset, different random seed etc.)? How would such an error distributions in the rollouts differ for baseline methods?

**Ethical Concerns:**

["NO or VERY MINOR ethics concerns only"]

**Final Justification:**

Authors have addressed my concerns (lack of error bars, additional concerns), however a stronger evaluation would be even better.

**Limitations:**

yes

**Quality:**

2

**Strengths And Weaknesses:**

*Strengths*
- This work addresses a significant and challenging problem in computational physics (nuclear fusion) where breakthroughs would have a large impact.
- The deep learning model demonstrates significant speedups compared to numerical solvers.

*Weaknesses*

- Ablations of the various model components are missing. What happens if the cross-attention modules, or latent integration modules, or the physics-informed loss terms are removed (and use simple squared error loss instead)?
- While UNets/CNN, ViTs and FNOs may not be scalable, they would still serve as baselines for small scale datasets. Particularly, it would be informative to see the quality ceiling that may be achieved using a non-physics informed deep learning approach. A simple MLP architecture is used, but that may be a weak baseline (also what are the parameter counts?)
- Even a simple baseline such as Gaussian process and MLPs achieve comparable quality on some datasets, even though they are missing several features such as physics informed learning dynamics. To me, the Table 2 appears too noisy to decide that GyroSwin is indeed the best performing model.

---

> ### Author Rebuttal · Authors · 2025-07-31
>
> We thank vuY7 for the time invested into the review and the constructive insights. We proceed to clarify the misconceptions and address the raised concerns.
>
> We noticed that the biggest critique is w.r.t. Table 2. This is in line with concerns raised by other reviewers (n9nG, WQGA, DBSK). From this feedback we distilled that we have failed to bring across the right message in Table 2. Our intention was to showcase that GyroSwin can indeed model turbulence in 5D fields `(2x32x8x16x85x32)` as opposed to widely used reduced numerical models or surrogates that predict only quantities resulting from turbulence. The baselines are well established regression tools which predict the flux directly from operation parameters ([R/Lt, R/Ln, s, q] = 4-valued vector). Furthermore, the quasilinear model is state-of-the-art and the most widely used reduced numerical model in practice [1,2]. The task of GyroSwin is multiple orders of magnitude harder, and thus a direct comparison to baselines might have been misleading. Most importantly, 3D and 5D phenomena are not captured by most of the baselines. We have now made this much clearer by:
> 1) We added conditioning to the flux prediction head of GyroSwin as all baselines are also conditioned on the operation parameters, which resulted in substantial gains in flux prediciton.
> 2) We added Pearson correlation coefficient for diagnostics and zonal flow evaluation in Table 2, including error bars for all evaluations to demonstrate the significance of our results.
> 3) We added additional baslines, like a simple Transformer model and provide results for scaling GyroSwin to larger parameter counts which results in significant gains.
> 4) We added information on the input shapes each method achieves and n/a fields to make it clear that certain evaluations are not applicable to the most baselines.
> 5) We added additional tables for runtime/memory/GFLOPs for different methodologies that could be applied to 5D input, but simply do not scale well, hence result in prohibitive cost.
> 6) We added a separate table for ablation studies for different prediction heads on a reduced data setup.
>
> ### Weaknesses
> __(1) Ablations of the various model components are missing. What happens if the cross-attention modules, or latent integration modules, or the physics-informed loss terms are removed (and use simple squared error loss instead)?__
>
> We agree with the reviewer that ablation studies are vital to determine crucial components of our model design. Therefore we report results on ablating the different heads added to the model for the different fields (5D distribution function, 3D potential fields, scalar flux). Importantly, ablating window attention is not feasible within the rebuttal period, as replacing it with full attention results in a drastic increase of computational burden (see Table in weakness (2)).  We conduct the ablations on a limited evaluation setup as extensive ablation studies on the whole dataset are infeasible due to the massive bulk of data that each variant is trained on (~2TB). We observe that the best OOD generalization performance is attained when combining all prediction heads.
>
> | Model | Params[M] | ID | OOD |
> | --- | --- | --- | --- |
> GyroSwin-OnlyFlux | 51.3 | 37.8 ± 9.18 | 54.38 ± 16.48 |
> GyroSwin-Onlydf | 48.4 | 132.45 ± 12.12 | 84.8 ± 13.69 |
> GyroSwin-df+phi | 94.7 | 41.16 ± 7.16 | 53.48 ± 15.91 |
> GyroSwin-All | 90.2 | 58.52 ± 14.68 | 38.52 ± 9.77 |
>
> __(2) While UNets/CNN, ViTs and FNOs may not be scalable, they would still serve as baselines for small scale datasets. Particularly, it would be informative to see the quality ceiling that may be achieved using a non-physics informed deep learning approach. What are the parameter counts?__
>
> It is true that other architectures such as CNNs/FNOs/ViTs can be extended to higher dimensions as well. However, these methods do not natively scale well to such high dimensional input. To demonstrate this, we report inference speed, peak memory consumption, GFLOPs, and parameter counts for those baselines in the table below. Numbers are reported on an NVIDIA H100 80GB HBM3, run on a single sample with resolution `(2, 32, 8, 16, 85, 32)`. FLOP counts are computed with the _fvcore_ library. F-CNN and F-FNO are rather small, with considerably less parameters than GyroSwin and ViT. This is because of memory constraints.
>
> It is important to note here that enabling these methods to efficiently operate at scale is not trivial, and probably requires a great deal of engineering, something that would arguably deserve it's own research work. This should both motivate the choice of our architecture and the reason why comparison with alternative neural networks is absent.
>
> Model|Inference [ms]|Memory [GB]|GFLOPs|Params [M]
> ---|---|---|---|---
> F-CNN|569.6|11.1|19408.7|2.0
> F-FNO|963.3|36.9|N/A|1.3
> ViT|87.7|1.2|1356.3|90.2
> ViT (Large)|1141.9|5.5|14316.3|936.8
> GyroSwin|30.4|5.0|756.0|90.2
> GyroSwin (Large)|262.4|11.4|9513.5|998.3
>
>
> Importantly, the dataset we compiled IS already _"small scale"_, as it is based on simplifying assumptions (single turbulence mode, adiabatic electrons, etc.). The spatial resolution used is among the smallest possible seen in gyrokinetics simulations, and that still results in around 100MB per time slice. For reference, higher fidelity simulations require resolutions that would amount to 10s of GBs per time slice. Therefore, running the suggested baselines even at our small scale is prohibitive. This is the reason for introducing GyroSwin, which effectively balances performance and efficiency. We will clarify this in our manuscript.
>
>
> __(3) Even a simple baseline such as Gaussian process and MLPs achieve comparable quality on some datasets, even though they are missing several features such as physics informed learning dynamics. To me, the Table 2 appears too noisy to decide that GyroSwin is indeed the best performing model.__
>
> Below we report the restructured version of Table 2 with the improved flux results, diagnostics and zonal flow predictions. _n/a_ means that the model is not capable of producing the result. We hope that the updated Table + updated experiments help to bring across our main point, i.e., **only GyroSwin can model turbulence**. Other methods can only predict quantities which arise from turbulence.
> We also added a vanilla Transformer baseline to model the sequential evolution of the flux trace.
>
> Method|Params|Input Shape|$Q$ ID|$Q$ OOD|$W(k_y)$|$Q_i(k_y)$|Zonal Flow
> ---|---|---|---|---|---|---|---
> GPR|6|(4,)|36.95±9.24|25.89±8.78|n/a|n/a|n/a
> MLP|37.3K|(4,)|59.52±15.01|29.59±11.19|n/a|n/a|n/a
> LSTM|5.3M|(10,)+(5,)|57.99±15.62|73.14±23.63|n/a|n/a|n/a
> Transformer|14.1|(10,)+(5,)|74.86±20.26|77.67±28.41|n/a|n/a|n/a|
> QL|1|(16,85,32)|40.74±7.47|53.3±14.77|0.04±0.15|0.54±0.46|n/a
> GyroSwin-Lin|51.3M|(32\,8,16,85,32)|38.03±10.73|31.19±8.73|n/a|n/a|n/a
> GyroSwin-S|90.2M|(32,8,16,85,32)|36.13±7.55|59.76±17.62|0.98±0.03|0.4±0.37|0.11±0.19
> GyroSwin-M|250.9M|(32,8,16,85,32)|36.15±7.45|76.24±16.99|0.98±0.03|0.3±0.37|0.15±0.22
> GyroSwin-L|998.3M|(32,8,16,85,32)|24.38±6.79|18.49±6.15|0.98±0.03|0.35±0.4|0.32±0.18
>
> We agree that Table 2 was indeed too noisy and not too informative to ultimately decide the best performing model. By adding the conditioning and reporting error bars we demonstrate that the results obtained with GyroSwin are indeed significant.
>
> ### Questions
> __(1) OOD Qi Performance: Why could explain why GyroSwin struggles in some OOD cases compared to GPR/MLP (Table 2), even though the modeling of GyroSwin has significantly more consideration for the underlying physics? I would have expected GyroSwin would overfit less.__
>
> The main reason for this was that the flux head was missing the conditioning on the 4 physical parameters, which is provided to all the baselines. Hence the comparison was inherently unfair. After conditioning the flux head via FiLM on those parameters, we observe vast improvements as reported in the table above.
>
> __(2a) While Fig 3 shows a successful rollout, would it be possible to visualize the distribution of errors over multiple rollouts as a function of time (different dataset, different random seed etc.)?__
>
> Thank you for that interesting suggestion. It is indeed possible to visualize the evolution of the error for GyroSwin across timesteps. Unfortunately, though, since GyroSwin models are expensive to train we do not yet have multiple models trained on multiple seeds at our disposal. However, we are planning to add such a visualization in the future.
>
> __(2b) How would such an error distributions in the rollouts differ for baseline methods?__
>
> Baseline methods such as MLP, GPR, or QL do not evolve over time, they predict average flux in the saturated state of the simulation. For the LSTM we observed that it tends to predict fluctuations close to the average of the distribution of fluxes. This is because the flux traces are very noisy and predicting the average is a strong local optima. We also observed that GyroSwin always predicts turbulent transport, even in the very beginning of the simulation where actually no turbulent transport occurs. We attribute this to the imbalanced nature of the dataset, as training samples of the saturated (turbulent) phase of the simulation are overrepresented.
>
> ### References
> [1] Kumar N. et al. Turbulent transport driven by kinetic ballooning modes in the inner core of jet hybrid h-modes. Nuclear Fusion.
>
> [2] Citrin J. et al. Tractable flux-driven temperature, density, and rotation profile evolution with the quasilinear gyrokinetic transport model QuaLiKiz.

---

> > ### Comment · Area_Chair_64EM · 2025-08-07
> >
> > Dear reviewer vuY7,
> >
> > As the deadline for author-reviewer is approaching (Aug 8), please comment if the authors adequately addresses your concerns, whether there are some concerns remaining, and update the score as appropriate. Please note that it is required to comment before acknowledgement.
> >
> > Best,
> > AC

---

> > ### Comment · Reviewer_vuY7 · 2025-08-08
> >
> > Thank you for the response. I'd encourage the authors to pursue this direction and invest in the engineering required to adapt existing methodologies (FNO, ViT) to 5-D Gyro simulations. Nevertheless, my concerns are satisfactorily addressed and I will raise my score accordingly.

---

### Official Review · Reviewer_386x · 2025-07-03

**Clarity:** 3
**Significance:** 3
**Originality:** 3
**Rating:** 5
**Confidence:** 2

**Summary:**

The paper presents a novel deep learning approach, GyroSwin, aimed at efficiently simulating nonlinear gyrokinetic plasma turbulence.

GyroSwin incorporates advanced techniques like latent cross-attention for interactions between 3D and 5D fields and channel-wise mode separation to isolate important nonlinear phenomena like zonal flows. It uses shifted window attention (Swin Transformer) to preserve locality and improve computational efficiency, allowing for accurate predictions in high-dimensional spaces. The model is trained using a multitask approach, predicting the 5D distribution function, 3D electrostatic potential, and heat flux.

**Questions:**

The proposed model structure seems to be much more complex than the baseline, such as MLP and LSTM. Can the author show them, as well as the parameter count and prediction speed of GyroSwin? This would be meaningful.

**Ethical Concerns:**

["NO or VERY MINOR ethics concerns only"]

**Final Justification:**

The author's ablation resolved my concerns. I think this manuscript does make a contribution because each component design seems to have a task related motivation, even though I am not an expert in plasma turbulence simulation. Thus, I keep my score.

**Limitations:**

The authors discuss the relevant limitations.

**Paper Formatting Concerns:**

No Formatting Concerns

**Quality:**

3

**Strengths And Weaknesses:**

**Strengths**

 1. This article is well written, with rich background descriptions and an overview of related work. Although I am not an expert in the field of controlled nuclear fusion, this introduction can help me quickly familiarize myself with the context.

2. Although the modules of the proposed surrogate model are existing, the design of each module is principled and strongly related to the task.

**Weaknesses**

It lacks, the ablation of each module, such as window attention, integral module, Multi-task training, etc., ablation experiments can better show the contribution of each module and inspire future work.

---

> ### Author Rebuttal · Authors · 2025-07-31
>
> Thank you for the positive evaluation of our submission, and the constructive feedback on how to improve it. We address the raised concerns as follows.
>
> ### Weaknesses
> __(1) It lacks, the ablation of each module, such as window attention, integral module, Multi-task training, etc., ablation experiments can better show the contribution of each module and inspire future work.__
>
> We agree with the reviewer that ablation studies are vital to determine crucial components of our model design. Therefore we report results on ablating the different heads added to the model for the different fields (5D distribution function, 3D potential fields, scalar flux). Importantly, ablating window attention is not feasible within the rebuttal period, as replacing it with full attention results in a drastic increase of computational burden (see Table below).
>
> Model|Inference [ms]|Memory [GB]|GFLOPs|Params [M]
> ---|---|---|---|---
> F-CNN|569.6|11.1|19408.7|2.0
> F-FNO|963.3|36.9|N/A|1.3
> ViT|87.7|1.2|1356.3|90.2
> ViT (Large)|1141.9|5.5|14316.3|936.8
> GyroSwin|30.4|5.0|756.0|90.2
> GyroSwin (Large)|262.4|11.4|9513.5|998.3
>
> We conduct the ablations on a limited evaluation setup as extensive ablation studies on the whole dataset are infeasible due to the massive bulk of data that each variant is trained on (~2TB). We observe that the best OOD generalization performance is attained when combining all prediction heads.
>
> | Model | Params[M] | ID | OOD |
> | --- | --- | --- | --- |
> GyroSwin-OnlyFlux | 51.3 | 37.8 ± 9.18 | 54.38 ± 16.48 |
> GyroSwin-Onlydf | 48.4 | 132.45 ± 12.12 | 84.8 ± 13.69 |
> GyroSwin-df+phi | 94.7 | 41.16 ± 7.16 | 53.48 ± 15.91 |
> GyroSwin-All | 90.2 | 58.52 ± 14.68 | 38.52 ± 9.77 |
>
> ### Questions
> __(1) The proposed model structure seems to be much more complex than the baseline, such as MLP and LSTM. Can the author show them, as well as the parameter count and prediction speed of GyroSwin? This would be meaningful.__
>
> It is true that GyroSwin is more complex than baselines such as MLP or LSTM, it is important to note though, that the task for MLP, GPR, and LSTM/Transformer is much simpler as they do not process 5D fields: __the input of GyroSwin is ~5 orders of magnitudes larger__: `(2, 32, 8, 16, 85, 32)` vs `(4,)` (or `(10,) + (4,)` for the sequence models, LSTM and Transformer). We realize that this has not been conveyed as desired in our submission.
>
> If we compare it with other architectures processing the same amount of data, the advantage of GyroSwin becomes more pronounced. Numbers are reported on an NVIDIA H100 80GB HBM3, run on a single sample with resolution `(2, 32, 8, 16, 85, 32)`. FLOP counts are computed with the _fvcore_ library.
>
> | Model | Input shape | Inference [ms] | Memory [GB] | GFLOPs | Params [M] |
> | --- | --- | --- | --- | --- | --- |
> | MLP | (4,) | 0.1 | <0.0 | <0.0 | <0.0 |
> | LSTM | (10,) + (5,) | 0.4 | 0.1 | 0.1 | 5.3 |
> | Transformer | (10,) + (5,) | 0.5 | 0.1 | 2.5 | 9.0 |
> | F-CNN | (2, 32, 8, 16, 85, 32) | 569.6 | 11.1 | 19408.7 | 2.0 |
> | F-FNO | (2, 32, 8, 16, 85, 32) | 963.3 | 36.9 | N/A | 1.3 |
> | ViT | (2, 32, 8, 16, 85, 32) | 87.7 | 1.2 | 1356.3 | 90.2 |
> | ViT (Large) | (2, 32, 8, 16, 85, 32) | 1141.9 | 5.5 | 14316.3 | 936.8 |
> | GyroSwin | (2, 32, 8, 16, 85, 32) | 30.4 | 5.0 | 756.0 | 90.2 |
> | GyroSwin (Large) | (2, 32, 8, 16, 85, 32) | 262.4 | 11.4 | 9513.5 | 998.3 |

---

> > ### Comment · Reviewer_386x · 2025-08-06
> >
> > The author's ablation resolved my concerns. I think this manuscript does make a contribution because each component design seems to have a task related motivation, even though I am not an expert in plasma turbulence simulation.

---

### Official Review · Reviewer_DBSK · 2025-07-05

**Clarity:** 2
**Significance:** 2
**Originality:** 3
**Rating:** 3
**Confidence:** 2

**Summary:**

The paper proposes an application-specific model to learn a type of PDE used for plasma simulation. GyroSwin builds a hierarchical Vision Transformer in 5D space and is trained on the 5D distribution function for the adiabatic electron approximation

**Questions:**

what is the model computational complexity and how does that compare to the baselines? It seems less clear in the manuscript.

**Ethical Concerns:**

["NO or VERY MINOR ethics concerns only"]

**Limitations:**

yes

**Paper Formatting Concerns:**

no formatting concerns

**Quality:**

3

**Strengths And Weaknesses:**

Strengths:
The approach appears sound. It makes use of vision transformer in 5D distribution function over time.
The paper explains the context well, and has demonstrated the significance of the problem in application contexts.

Weaknesses:
The paper can benefit from discussion of related works in neural approaches to solving Gyrokinetics problems, and their weaknesses compared to the proposed model.

In table 2, standard deviation needs to be reported, to assess significance of the results. Furthermore, a transformer-based methods could potentially help readers judge the advantages of proposed methods. LSTM methods might be less effective in large-scale training. Other advanced methods such as state-space methods could be considered as well.

The paper lacks ablation studies.

The paper includes a lot of domain-specific details for the application, not commonly encountered in the deep learning community. Unfortunately I lack the physics background to assess the relevance and importance of the proposed simulation model.

---

> ### Author Rebuttal · Authors · 2025-07-31
>
> We'd first like to thank DBSK for their review and their constructive feedback.
>
> We noticed that one of the biggest critiques is w.r.t. Table 2. This is in line with concerns raised by other reviewers (n9nG, WQGA, vuY7). From this feedback we distilled that we have failed to bring across the right message in Table 2. Our intention was to showcase that GyroSwin can indeed model turbulence in 5D fields `(2x32x8x16x85x32)` as opposed to widely used reduced numerical models or surrogates that predict only quantities resulting from turbulence. The baselines are well established regression tools which predict the flux directly from operation parameters ([R/Lt, R/Ln, s, q] = 4-valued vector). Furthermore, the quasilinear model is state-of-the-art and the most widely used reduced numerical model in practice [1,2]. The task of GyroSwin is multiple orders of magnitude harder, and thus a direct comparison to baselines might have been misleading. Most importantly, 3D and 5D phenomena are not captured by most of the baselines. We have now made this much clearer by:
> 1) We added conditioning to the flux prediction head of GyroSwin as all baselines are also conditioned on the operation parameters, which resulted in substantial gains in flux prediciton.
> 2) We added Pearson correlation coefficient for diagnostics and zonal flow evaluation in Table 2, including error bars for all evaluations to demonstrate the significance of our results.
> 3) We added additional baslines, like a simple Transformer model and provide results for scaling GyroSwin to larger parameter counts which results in significant gains.
> 4) We added information on the input shapes each method achieves and n/a fields to make it clear that certain evaluations are not applicable to the most baselines.
> 5) We added additional tables for runtime/memory/GFLOPs for different methodologies that could be applied to 5D input, but simply do not scale well, hence result in prohibitive cost.
> 6) We added a separate table for ablation studies for different prediction heads on a reduced data setup.
>
>
> ### Weaknesses
> __(1) The paper can benefit from discussion of related works in neural approaches to solving Gyrokinetics problems, and their weaknesses compared to the proposed model.__
>
> For additional information and proper referencing, we would like to point the reviewer to lines 123 to 137 in the Related Works section.
> Literature on machine learning approaches for gyrokinetics is rather sparse. Most works focus on predicting average fluxes computed via reduced quasilinear models from physical parameters, employing traditional machine learning methods like gaussian processes or basic neural techniques such as MLPs. More recently there have been a few attempts to learn nonlinear gyrokinetics through reduced and integrated quantities, such as wavenumbers and slices of the electrostatic potential field. Note that these methods are limited to scalars, 1D and 2D information, so they cannot properly model the gyrokinetics nonlinear behaviour. This distinguishes GyroSwin from any previous work on this topic.
>
> __(2) In table 2, standard deviation needs to be reported, to assess significance of the results. Furthermore, a transformer-based methods could potentially help readers judge the advantages of proposed methods. LSTM methods might be less effective in large-scale training. Other advanced methods such as state-space methods could be considered as well.__
>
> Below we report the restructured version of Table 2 with the improved flux results, diagnostics and zonal flow predictions. _n/a_ means that the model is not capable of producing the result. We hope that the updated Table + updated experiments help to bring across our main point, i.e., **only GyroSwin can model turbulence**. Other methods can only predict quantities which arise from turbulence.
> We also added a vanilla Transformer baseline to model the sequential evolution of the flux trace.
> We agree that state-space models could be added as well, but our main point is that we require specialized architectures like GyroSwin that scale well to process the 5D data which allows to actively model turbulence instead of just predicting quantities that result from it.
>
> Method|Params|Input Shape|$Q$ ID↓|$Q$ OOD↓|$W(k_y)$↑|$Q_i(k_y)$↑|Zonal Flow↑
> ---|---|---|---|---|---|---|---
> GPR|6|(4,)|36.95±9.24|25.89±8.78|n/a|n/a|n/a
> MLP|37.3K|(4,)|59.52±15.01|29.59±11.19|n/a|n/a|n/a
> LSTM|5.3M|(10,)+(5,)|57.99±15.62|73.14±23.63|n/a|n/a|n/a
> Transformer|14.1|(10,)+(5,)|74.86±20.26|77.67±28.41|n/a|n/a|n/a|
> QL|1|(16,85,32)|40.74±7.47|53.3±14.77|0.04±0.15|0.54±0.46|n/a
> GyroSwin-Lin|51.3M|(32\,8,16,85,32)|38.03±10.73|31.19±8.73|n/a|n/a|n/a
> GyroSwin-S|90.2M|(32,8,16,85,32)|36.13±7.55|59.76±17.62|0.98±0.03|0.4±0.37|0.11±0.19
> GyroSwin-M|250.9M|(32,8,16,85,32)|36.15±7.45|76.24±16.99|0.98±0.03|0.3±0.37|0.15±0.22
> GyroSwin-L|998.3M|(32,8,16,85,32)|24.38±6.79|18.49±6.15|0.98±0.03|0.35±0.4|0.32±0.18
>
> __(3) The paper lacks ablation studies.__
>
> We agree with the reviewer that ablation studies are vital to determine crucial components of our model design. Therefore we report results on ablating the different heads added to the model for the different fields (5D distribution function, 3D potential fields, scalar flux). Importantly, ablating window attention is not feasible within the rebuttal period, as replacing it with full attention results in a drastic increase of computational burden (see Table in Question (1)).  We conduct the ablations on a limited evaluation setup as extensive ablation studies on the whole dataset are infeasible due to the massive bulk of data that each variant is trained on (~2TB). We observe that the best OOD generalization performance is attained when combining all prediction heads.
>
> | Model | Params[M] | ID | OOD |
> | --- | --- | --- | --- |
> GyroSwin-OnlyFlux | 51.3 | 37.8 ± 9.18 | 54.38 ± 16.48 |
> GyroSwin-Onlydf | 48.4 | 132.45 ± 12.12 | 84.8 ± 13.69 |
> GyroSwin-df+phi | 94.7 | 41.16 ± 7.16 | 53.48 ± 15.91 |
> GyroSwin-All | 90.2 | 58.52 ± 14.68 | 38.52 ± 9.77 |
>
> ### Questions
> __(1) what is the model computational complexity and how does that compare to the baselines?__
>
> Theoretical computational complexity of 5D swin layers, with an input of resolution $v \times m \times s \times x \times y$ with $C$ channels and window size $W$ (assumed square for simplicity) is $O(\text{W-MSA}) = 4 \: vmsxy \: C^2 + 2W^5 \: vmsxy \: C$
>
> making it linear in the input resolution.
>
> In practice, we measure inference time, memory and GFLOPs for our models and compare with commonly used architectures (factorized CNN and factorized FNO). Numbers are reported on an NVIDIA H100 80GB HBM3, run on a single sample with resolution `(2, 32, 8, 16, 85, 32)`. FLOP counts are computed with the _fvcore_ library.
>
> Model | Inference [ms] | Memory [GB] | GFLOPs | Params [M]
> --- | --- | --- | --- | ---
> F-CNN | 569.6 | 11.1 | 19408.7 | 2.0
> F-FNO | 963.3 | 36.9 | N/A | 1.3
> GyroViT  | 87.7 | 1.2 | 1356.3 | 90.2
> GyroViT (Large) | 1141.9 | 5.5 | 14316.3 | 936.8
> GyroSwin | 30.4 | 5.0 | 756.0 | 90.2
> GyroSwin (Large) | 262.4 | 11.4 | 9513.5 | 998.3
>
> When processing the entire 5D fields, GyroSwin is considerably faster and less memory demanding than even small and simple convolution-based models. However, if we compare to the baselines that we originally reported in Table 2, GyroSwin is obviously much more expensive since __the input of GyroSwin is ~5 orders of magnitudes larger__: `(2, 32, 8, 16, 85, 32)` vs `(4,)` (or `(10,) + (4,)` for the sequence models, LSTM and Transformer).
> Nonetheless, we report numbers for the simple baselines below.
>
> | Model | Inference [ms] | Memory [GB] | GFLOPs | Params [M] |
> | --- | --- | --- | --- | --- |
> | MLP | 0.1 | <0.0 | <0.0 | <0.0 |
> | LSTM | 0.4 | 0.1 | 0.1 | 5.3 |
> | Transformer | 0.5 | 0.1 | 2.5 | 9.0 |
>
>
> ### References
> [1] Kumar N. et al. Turbulent transport driven by kinetic ballooning modes in the inner core of jet hybrid h-modes. Nuclear Fusion.
>
> [2] Citrin J. et al. Tractable flux-driven temperature, density, and rotation profile evolution with the quasilinear gyrokinetic transport model QuaLiKiz.

---

### Official Review · Reviewer_WQGA · 2025-07-14

**Clarity:** 4
**Significance:** 3
**Originality:** 2
**Rating:** 4
**Confidence:** 4

**Summary:**

In magnetic confinement fusion, plasma turbulence is a critical phenomenon that can result in heat and particle transport towards the reactor walls. However, this phenomenon is based on a 5D PDE that is computationally intensive. While quasi-linear approximation approaches neglect some nonlinear physics that are crucial to modeling the turbulence. Therefore, the authors propose GyroSwin, a multitask hierarchical ViT to process the 5D data to capture nonlinear dynamics.

**Questions:**

(1) What is W-MSA and SW-MSA in Figure 2? \
(2) Could you provide more insights into how cross-attention is related to 5D-3D integral? \
(3) In Table 2, traditional GPR and MLP approaches seem to excel at in-distribution and out-of-distribution scenarios. Is it because the problem  (5D->0D) is not sophisticated enough? \
(4) In Figure 3, why QL, GPR, and MLP results are flat lines? Is it possible to use these models to predict a trend like what GyroSwin does? \
(5) Is it possible to use traditional approaches (e.g., MLP) to obtain 5D->1D evaluation results and show them together in Figure 4? For example, taking 5D data as input and getting 1D data as output?

**Ethical Concerns:**

["NO or VERY MINOR ethics concerns only"]

**Limitations:**

yes

**Quality:**

3

**Strengths And Weaknesses:**

**Strengths:**\
(1) The motivation of this work is clear. \
(2) The model design goals are reasonable and clearly listed in Section 3. \
(3) The inclusion of physics could be inspiring to researchers on AI for science, especially in terms of high-dimensional PDEs. \
\
**Weaknesses:**\
(1) Extending model capabilities to support higher dimensions is not entirely new. \
(2) The concept of multitask prediction is a well-established technique in AI. \
(3) As an application-specific approach, the results in Table 2 are outperformed by generic and relatively simple approaches in many cases. \
(4) Some acronyms are not properly defined.

---

> ### Author Rebuttal · Authors · 2025-07-31
>
> We sincerely thank the reviewer for their comments. We address them in the following subsections.
>
> We noticed that one of the biggest critiques is wrt Tab 2, in line with concerns raised by reviewers (n9nG, DBSK, vuY7). From this we distilled that we have failed to bring across the intended message, i.e., to show that GyroSwin can indeed model turbulence in 5D fields `(2x32x8x16x85x32)` as opposed to widely used reduced numerical models or surrogates that predict only quantities resulting from turbulence. The baselines are well established regression tools which predict flux directly from operation parameters ([R/Lt, R/Ln, s, q] = 4 scalars). Furthermore, the quasilinear model is SOTA and the most widely used reduced numerical model in practice [1,2]. The task of GyroSwin is considerably harder, and a direct comparison to baselines was misleading. __3D and 5D phenomena are not captured by most of the baselines__. We make this clearer by adding:
> 1) Conditioning the flux prediction head of GyroSwin, which resulted in substantial gains in flux prediciton.
> 2) Pearson correlation coefficient for diagnostics and zonal flow evaluation in Table 2, including error bars for all evaluations to demonstrate the significance of our results.
> 3) Additional baselines, (flux sequence Transformer); scaling GyroSwin to larger sizes.
> 4) Information on the input shapes for each method; diagnostic fields to clarify that certain evaluations are not applicable to the most baselines.
> 5) Additional tables for runtime/memory/GFLOPs for different methodologies that could be applied to 5D input but do not scale well and result in prohibitive cost.
> 6) A separate table ablating different heads on a reduced data setup.
>
> ### Weaknesses
> __(1) Extending model capabilities to support higher dimensions is not entirely new.__
>
> It is true that other architectures have been extended to higher dimensions. E.g., Convolutions and FNOs have factorized implementations ([3, 4]), making them in theory generalizable to n-dimensions. Also, vanilla ViTs can in principle be applied to nD data, similarly to how we apply Swin. However, these methods do not natively scale well to higher dim inputs as we show in the table below.
>
> Model|Inference [ms]|Memory [GB]|GFLOPs|Params [M]
> ---|---|---|---|---
> F-CNN|569.6|11.1|19408.7|2.0
> F-FNO|963.3|36.9|N/A|1.3
> ViT|87.7|1.2|1356.3|90.2
> GyroSwin|30.4|5.0|756.0|90.2
>
> This highlights the need for more scalability.
> Hierarchical ViTs such as Swin [5] have been extended to Videos (3D [6]) and medical images (4D [7]), but not beyond that. We generalize Swin by implementing n-dimenional _local convolutions_, used in patching/merging/expansion layers and in window attention. To the best of our knowledge, this is novel.
>
> Furthermore, we incorporate inductive-biases, namely channelwise mode separation and latent integration modules.
> These are tailored to our application domain, but are simple ways of including inductive biases into model design, which may be relevant to the community.
>
> __(2) The concept of multitask prediction is a well-established technique in AI.__
>
> This is absolutely true, we did not claim that the multitask setup is one of our contributions. It is simply a tool to enhance our performance.
>
> __(3) As an application-specific approach, the results in Table 2 are outperformed by generic and relatively simple approaches in many cases.__
>
> As mentioned above, the main cause of the result inconsistency at submission was that the flux head was not conditioned on operational parameters.
>
> Below we report the restructured Table 2 with improved flux results, diagnostics and zonal flow predictions. _n/a_ means that the model is incapable of producing the result. We hope that the updated Table + updated experiments help to bring across our main point, i.e., **only GyroSwin can model turbulence**. Others can only predict quantities which arise from turbulence.
>
> Method|Params|Input Shape|$Q$ ID↓|$Q$ OOD↓|$W(k_y)$↑|$Q_i(k_y)$↑|Zonal Flow↑
> ---|---|---|---|---|---|---|---
> GPR|6|(4,)|36.95±9.24|25.89±8.78|n/a|n/a|n/a
> MLP|37.3K|(4,)|59.52±15.01|29.59±11.19|n/a|n/a|n/a
> LSTM|5.3M|(10,)+(5,)|57.99±15.62|73.14±23.63|n/a|n/a|n/a
> Transformer|14.1|(10,)+(5,)|74.86±20.26|77.67±28.41|n/a|n/a|n/a|
> QL|1|(16,85,32)|40.74±7.47|53.3±14.77|0.04±0.15|0.54±0.46|n/a
> GyroSwin-Lin|51.3M|(32\,8,16,85,32)|38.03±10.73|31.19±8.73|n/a|n/a|n/a
> GyroSwin-S|90.2M|(32,8,16,85,32)|36.13±7.55|59.76±17.62|0.98±0.03|0.4±0.37|0.11±0.19
> GyroSwin-M|250.9M|(32,8,16,85,32)|36.15±7.45|76.24±16.99|0.98±0.03|0.3±0.37|0.15±0.22
> GyroSwin-L|998.3M|(32,8,16,85,32)|24.38±6.79|18.49±6.15|0.98±0.03|0.35±0.4|0.32±0.18
>
> __(4) Some acronyms are not properly defined.__
> We address this point in question (1).
>
> ### Questions
>
> __(1) What is W-MSA and SW-MSA in Figure 2?__
> Thanks for noticing this. These are indeed not defined in the paper, and they were borrowed from the original swin paper. We now formalize those as they are used in our setup.
>
> - W-MSA is Window Multihead Self Attention (MSA). It uses the even-layer window partition scheme, and performs self-attention only within the windows (in parallel). As a slightly lax formulation, $\text{W-MSA}(x)=\left[\sigma\left( \frac{(x_w W_h^q)(x_w W_h^k)^\top}{\sqrt{d}} + B \right) (x_w W_h^v)\right]_{h \in \text{heads}}W^o$, where $x_w$ is the window partitioned input, with $x_w.\texttt{shape} = \texttt{(batch*nwindows, w0, w1, ..., channels)}$, and $W_h^q, W_h^k, W_h^v, W_h^o$ are the q, k, v and out projection per-head $h$ projections of self-attention, and $B$ is a bias which encodes relative positional information.
>
> - SW-MSA is Shifted Window MSA, it uses the shifted, odd-layer partition scheme. This introduces some information exchange with the neighbors. $\text{SW-MSA}(x)=\texttt{roll}(\text{W-MSA}(\texttt{roll}(x, −W/2)), +W/2)$, where $W$ is the window size and $\texttt{roll}$ is a matrix roll operation, which circularly shifts the input along the spatial dimensions by a given displacement (equivalent to `np.roll`).
>
> Please refer to the 2021 Shifted Window Transformer paper [5] for simpler 2D visualizations.
>
> __(2) More insights into how cross-attention is related to 5D-3D integral?__
>
> The inspiration for the 5D-3D mix and the $\int_{V_{space}}$ modules comes from the integral expressed in Eq. 2. It consists of a velocity space integral (to obtain potential fields) and a spatial integral (to obtain fluxes).
>
> First, 3D latents of the electrostatic potentials are computed (starting from 5D latents) with the $\int_{V_{space}}$ block. This uses cross-attention, with a learned scalar query which reduces the velocity space of the 5D key-value, (akin to perceiver-pooling [8]). The output of this operation is a latent 3D field.
>
> Then, the 5D-3D mix module is used to allow exchange of information at different scales. Here, cross-attention is performed between the latents of the 3D potential and 5D distribution function (connection expressed by the $v_\chi$ term in Eq. 2). The output is another 3D latent field.
>
> Finally, this 3D latent is used both for aiding potential reconstruction (3D potential fields) and for flux estimation (scalar). Fluxes are obtained by pooling the latents obtained with 5D-3D mixing spatially, and finally applying an MLP that predicts a scalar value.
>
> __(3) In Tab2, traditional GPR and MLP approaches seem to excel at ID and OOD scenarios. Is the problem (5D->0D) not sophisticated enough?__
>
> This is a very legitimate question, because the terminology of “5D->0D” evaluation for Table 2 is partially inaccurate. In fact, only GyroSwin treats it as an actual “5D->0D” problem, while all other baselines do not and cannot process 5D data at all. __Refer to weakness (3) for full answer.__
>
> __(4) In Figure 3, why QL, GPR, and MLP results are flat lines?__
>
> Thank you for pointing out that this isn't clear.
> Generally, it is __not__ possible to extend GPR and QL to predict trend over time. The target for __QL, GPR and MLP is the time-averaged heat flux__ $\bar{Q}$ in the statistically stable (saturated) phase. These models do not have the concept of time, and are trained to regress directly from physical parameters (GPR, MLP) or simplified linear equations (QL) to the saturated average flux $\bar{Q}$. While theoretically it would be possible to extend MLP to predict flux over time by adding timestep information, we expect this to not perform better than LSTM/Transformer due to its lack of expressivenes. Due to its autoregressive nature, GyroSwin is able to reproduce the evolution of turbulent flux into the saturated phase after a certain number of steps.
>
> __(5) Is it possible to use traditional approaches (e.g., MLP) to obtain 5D->1D evaluation results and show them together in Figure 4?__
>
> Yes, theoretically it is possible, but it would require flattening out the 5D field into a vector. Assuming a hidden size of the MLP of 512, a single weight matrix would carry over 10B parameters. Since 5D->1D evaluation requires a predicted 5D field, two of those are needed. The flattening operation destroys all locality in the field, which is of utmost importance.
>
> ### References
> [1] Kumar N. et al. Turbulent transport driven by kinetic ballooning modes in the inner core of jet hybrid h-modes. Nuclear Fusion.
>
> [2] Citrin J. et al. Tractable flux-driven temperature, density, and rotation profile evolution with the quasilinear gyrokinetic transport model QuaLiKiz.
>
> [3] Wang M. et al. (2017). Factorized Convolutional Neural Networks. *ICCV*.
>
> [4] Tran A. et al. (2023). Factorized Fourier Neural Operators. *ICLR*.
>
> [5] Liu Z. et al. (2021). Swin Transformer: Hierarchical Vision Transformer using Shifted Windows. *ICCV*.
>
> [6] Liu Z. et al. (2021). Video Swin Transformer.
>
> [7] Kim P. et al., Moon T. (2023). SwiFT: Swin 4D fMRI Transformer. *NeurIPS*.
>
> [8] Alkin B. et al. (2024). Universal Physics Transformers: A Framework For Efficiently Scaling Neural Operators. *NeurIPS*.

---

### Official Review · Reviewer_n9nG · 2025-07-15

**Clarity:** 3
**Significance:** 3
**Originality:** 3
**Rating:** 3
**Confidence:** 4

**Summary:**

This paper presents GyroSwin, a deep learning surrogate model for 5D gyrokinetic plasma turbulence simulations; a critical and computationally demanding problem in fusion research. Leveraging a hierarchical Swin Transformer architecture adapted for 5D data, GyroSwin captures key nonlinear effects, including zonal flows, that are often missed by traditional reduced models. GyroSwin predicts the 5D distribution function, 3D electrostatic potential, and scalar fluxes, achieving accuracy close to high-fidelity simulations while operating orders of magnitude faster. The approach enables new capabilities for data-driven plasma modeling.

**Questions:**

- In Table 2, GyroSwin does not consistently outperform classical baselines such as GPR, MLP, and LSTM, both for in-distribution and out-of-distribution data. Can the authors comment on the scenarios where GyroSwin underperforms, and explain why it does not achieve better accuracy in these cases?

- The paper lacks theoretical analysis and does not provide ablation studies. Could the authors include more theoretical justification and perform ablation studies to better understand the contributions of each module?

- The efficiency comparison is not entirely fair, as the numerical solver is evaluated on CPU while GyroSwin uses an A100 GPU. Could the authors provide a more balanced speed comparison, and also report the training cost, number of parameters, and GFLOPs for each model in Table 2?

- The paper does not compare GyroSwin with recently developed numerical methods or state-of-the-art neural network models. Could the authors expand their experimental comparison to include such baselines for a fairer assessment?

**Ethical Concerns:**

["NO or VERY MINOR ethics concerns only"]

**Final Justification:**

The authors have made an effort to provide a more detailed explanation of the experimental results in the rebuttal. However, the baseline models appear to be weak baselines, and despite the large difference in the number of parameters, the results do not show a substantial improvement. As these main concerns remain unresolved, I will maintain my original borderline reject score.

**Limitations:**

Yes.

**Quality:**

3

**Strengths And Weaknesses:**

Strength:
- The paper is clearly written and well organized, making complex concepts accessible to the reader. Also, the work addresses a significant and challenging problem in fusion plasma physics, which is accurate and efficient modeling of 5D gyrokinetic turbulence.

- GyroSwin introduces a multitask hierarchical Vision Transformer framework capable of processing high-dimensional scientific data. The approach leverages channel-wise mode separation, latent cross-attention across fields of varying dimensionality, and novel integration layers.

- The method simultaneously predicts the 5D distribution function, 3D electrostatic potential, and scalar fluxes, and demonstrates improved accuracy in flux prediction and nonlinear dynamics compared to reduced numerical models.

Weaknesses:
- In the main results table (Table 2), GyroSwin does not consistently outperform other methods. In fact, it often fails to achieve better results than classical baselines such as GPR, MLP, and LSTM, both in in-distribution (ID) and out-of-distribution (OOD) settings.

- The paper lacks theoretical analysis or evidence to explain why the proposed approach should work well. There is also no ablation study or detailed investigation into the factors driving the model’s performance.

- The efficiency comparison is not entirely fair: the numerical solver runs on CPU while GyroSwin is evaluated on an A100 GPU. A direct speed comparison under these conditions is questionable. Additionally, training cost comparisons with other models are missing; it would be helpful to report the number of parameters and GFLOPs for each method. Given the scale of the architecture, the training cost for GyroSwin also appears to be quite high.

- The paper lacks sufficient comparison with other methods, especially recently developed numerical approaches and neural network-based models. For a fair evaluation, it would be important to benchmark GyroSwin against both state-of-the-art numerical solvers and the latest deep learning models. In general, it is difficult for surrogate models to surpass the accuracy of modern numerical methods and advanced neural architectures.

---

> ### Author Rebuttal · Authors · 2025-07-31
>
> We appreciate the constructive feedback, recognizing both our technical contributions (accurate and efficient modeling of 5D gyrokinetic turbulence, multitask hierarchical ViT framework) and opportunities to strengthen the presentation of our work.
>
> We noticed that the biggest critique is wrt Tab 2, in line with concerns raised by other reviewers (WQGA, DBSK, vuY7). From this we distilled that we have failed to bring across the right message in Table 2. Our intention was to showcase that GyroSwin can indeed model turbulence in 5D fields `(2x32x8x16x85x32)` as opposed to widely used reduced numerical models or surrogates that predict only quantities resulting from turbulence. The baselines are well established regression tools which predict the flux directly from operation parameters ([R/Lt, R/Ln, s, q] = 4-valued vector). Furthermore, the quasilinear model is state-of-the-art and the most widely used reduced numerical model in practice [1,2]. The task of GyroSwin is multiple orders of magnitude harder, and thus a direct comparison to baselines might have been misleading. Most importantly, 3D and 5D phenomena are not captured by most of the baselines. We have now made this much clearer by:
> 1) Added conditioning to the flux prediction head of GyroSwin as all baselines are also conditioned on the operation parameters, which resulted in substantial gains in flux prediciton.
> 2) Pearson correlation coefficient for diagnostics and zonal flow evaluation in Table 2, including error bars for all evaluations to demonstrate the significance of our results.
> 3) Additional baslines, like a simple Transformer model and provide results for scaling GyroSwin to larger parameter counts which results in significant gains.
> 4) Adding information on the input shapes each method achieves and n/a fields to make it clear that certain evaluations are not applicable to the most baselines.
> 5) Additional tables for runtime/memory/GFLOPs for different methodologies that could be applied to 5D input, but simply do not scale well, hence result in prohibitive cost.
> 6) A separate table for ablation studies for different prediction heads on a reduced data setup.
>
> ### Weaknesses
> __(1) GyroSwin does not consistently outperform other methods (Table 2).__
>
> As mentioned above, the main reason for the inconsistency in the results at the time of submission was that the flux prediction head was not conditioned on the operational parameters.
> Below we report the restructured version of Table 2 with the improved flux results, diagnostics and zonal flow predictions. _n/a_ means that the model is not capable of producing the result. We hope that the updated Table + updated experiments help to bring across our main point, i.e., **only GyroSwin can model turbulence**. Other methods can only predict quantities which arise from turbulence.
>
> Method|Params|Input Shape|$Q$ ID↓|$Q$ OOD↓|$W(k_y)$↑|$Q_i(k_y)$↑|Zonal Flow↑
> ---|---|---|---|---|---|---|---
> GPR|6|(4,)|36.95±9.24|25.89±8.78|n/a|n/a|n/a
> MLP|37.3K|(4,)|59.52±15.01|29.59±11.19|n/a|n/a|n/a
> LSTM|5.3M|(10,)+(5,)|57.99±15.62|73.14±23.63|n/a|n/a|n/a
> Transformer|14.1|(10,)+(5,)|74.86±20.26|77.67±28.41|n/a|n/a|n/a|
> QL|1|(16,85,32)|40.74±7.47|53.3±14.77|0.04±0.15|0.54±0.46|n/a
> GyroSwin-Lin|51.3M|(32\,8,16,85,32)|38.03±10.73|31.19±8.73|n/a|n/a|n/a
> GyroSwin-S|90.2M|(32,8,16,85,32)|36.13±7.55|59.76±17.62|0.98±0.03|0.4±0.37|0.11±0.19
> GyroSwin-M|250.9M|(32,8,16,85,32)|36.15±7.45|76.24±16.99|0.98±0.03|0.3±0.37|0.15±0.22
> GyroSwin-L|998.3M|(32,8,16,85,32)|24.38±6.79|18.49±6.15|0.98±0.03|0.35±0.4|0.32±0.18
>
> __(2) Lacks theoretical analysis why the proposed approach should work well.__
>
> We appreciate the reviewer’s request for theoretical justification. However, a general theoretical framework for selecting optimal surrogate models in high-dimensional nonlinear gyrokinetics does not currently exist. Our results show that GyroSwin successfully generalizes over ID and OOD evaluation scenarios and accurately captures some of the underlying physics. This is the strongest form of validation that is currently feasible. Our work can be viewed as an initial step towards establishing practical surrogate modeling techniques in this domain, and we hope it will motivate future theoretical study.
>
> __(3) Efficiency comparison is not entirely fair.__
>
> We acknowledge that a comparison purely based on clocktime between CPU and GPU is not fair. A better comparison might be with floating-point operation count betwen the two simulations: to produce a single 5D dump, the numerical code GKW takes between 20-80 seconds on 64 CPU cores, with the theoretical peak GFLOP is up to 4200 GFLOPs (double precision, Ice Lake 4.1GHz 76 cores CPU, 2 FMA x 8 DP = 16 FLOPs/cycle). For GyroSwin, the runtime (NVIDIA H100 80GB HBM3) is ~30ms 756 GFLOPs. More details on the machine learning side is in the table in point (5).
>
> In a practical scenario of full integrated plasma simulations, it is typically necessary to compute turbulent heat fluxes multiple times, with up to thousands of simulation runs (at multiple radial locations and at regular intervals). Additionally, when adding more complex phenomena to the system, the numerical simulation will increase computation time substantially while surrogate models scale more favorably.
>
> __(4) Training cost comparisons with other models are missing.__
>
> We report training cost in terms of GFLOPS, number of parameters and peak memory usage during training for each model in the table below. GFLOPs are computed as the number of updates times the GFLOPs for a single forward pass times two (considering forward and backward pass). Importantly, though, as outlined above, the increase in GFLOPs comes from the fact that GyroSwin processes the whole 5D field, whereas the baselines merely process vectors of scalars.
>
> Model|Memory[GB]|GFLOPs|Params[M]
> ---|---|---|---
> MLP|0.002|6.1|<0.0
> LSTM|2.7|600|5.3
> Transformer|6.4|15K|9.0
> GyroSwin|26.6|263.8G|90.2
> GyroSwin(Large)|60.9|3.3B|998.3
>
> __(5) Lacks sufficient comparison with other methods.__
>
> Unfortunately, as far as we know, there are no other neural networks that can reliably be trained on the same task, scale and setup as GyroSwin. On one hand there is very limited research in general to extending neural networks beyond 4D- inputs, and on the other even when common layers such as convolution are implemented in 5D, they struggle to scale to higher resolutions due to the shear data size. To drive the second point, the following table compares very basic (factorized) CNN and FNO models against GyroSwin when applied to a single data sample of shape `(2, 32, 8, 16, 85, 32)`.
>
> Model|Inference[ms]|Memory[GB]|GFLOPs|Params[M]
> ---|---|---|---|---
> F-CNN|569.6|11.1|19408.7|2.0
> F-FNO|963.3|36.9|N/A|1.3
> ViT|87.7|1.2|1356.3|90.2
> ViT-L|1141.9|5.5|14316.3|936.8
> GyroSwin-S|30.4|5.0|756.0|90.2
> GyroSwin-L|262.4|11.4|9513.5|998.3
>
> Scaling other popular methods such as modern conv-UNet and FNO to our task is not trivial, and probably requires a great deal of engineering, something that would arguably deserve it's own research work. This should both motivate the choice of our architecture and the reason why comparison with alternative neural networks is absent.
>
> As for other numerical approaches, the aim of GyroSwin is not outperform sota numerical methods on accuracy. Our goal is to speed up the computation of quantities of interest and enable practictioners to afford more runs for the same compute time. The numerical code we employ to generate data, GKW, is considered state of the art in computational plasma physics [3].
>
> Finally, GyroSwin is based on ViTs, in particular shifted window attention and special cross-attention layers, which are arguably among the latest deep learning techniques.
>
> ### Questions
> __(1) In Table 2, GyroSwin does not consistently outperform classical baselines such as GPR, MLP, and LSTM, both for ID and OOD data. Can the authors comment on the scenarios where GyroSwin underperforms, and explain why it does not achieve better accuracy in these cases?__
>
> This is discussed in the answer to weakness (1).
>
> __(1) Ablation studies to better understand the contributions of each module.__
>
> We agree with the reviewer that ablation studies are vital to determine crucial components of our model design. Therefore we report results on ablating the different heads added to the model for the different fields (5D distribution function, 3D potential fields, scalar flux). Importantly, ablating window attention is not feasible within the rebuttal period, as replacing it with full attention results in a drastic increase of computational burden (see Table in weakness (5)).  We conduct the ablations on a limited evaluation setup as extensive ablation studies on the whole dataset are infeasible due to the massive bulk of data that each variant is trained on (~2TB). We observe that the best OOD generalization performance is attained when combining all prediction heads.
>
> Model|Params[M]|ID|OOD
> ---|---|---|---
> GyroSwin-OnlyFlux|51.3|37.8±9.18|54.38±16.48
> GyroSwin-Onlydf|48.4|132.45±12.12|84.8±13.69
> GyroSwin-df+phi|94.7|41.16±7.16|53.48±15.91
> GyroSwin-All|90.2|58.52±14.68|38.52±9.77
>
> __(2) Training cost, number of parameters, and GFLOPs for each model.__
>
> See response to weakness (4).
>
> __(3) The paper does not compare GyroSwin with recently developed numerical methods or state-of-the-art neural network models. Could the authors expand their experimental comparison to include such baselines for a fairer assessment?__
>
> See response to weakness (5).
>
> ### References
> [1] Kumar N. et al. Turbulent transport driven by kinetic ballooning modes in the inner core of jet hybrid h-modes. Nuclear Fusion.
>
> [2] Citrin J. et al. Tractable flux-driven temperature, density, and rotation profile evolution with the quasilinear gyrokinetic transport model QuaLiKiz.
>
> [3] Najlaoui A. et al. Verifying turbulence model reduction in high β tokamak plasmas (2025) *Plasma Phys. Control*

---

> > ### Comment · Reviewer_n9nG · 2025-08-05
> >
> > Thank you for your revision. However, even in the updated table, the comparisons are still limited to only basic models, which I consider to be weak baselines. Furthermore, despite the large difference in the number of parameters, the results do not show a significant improvement. I will therefore maintain my original score.

---

> > > ### Author Response · Authors · 2025-08-05
> > >
> > > Thank you for your engagement in a discussion.
> > > As pointed out in the rebuttal (Weakness 5), commonly used models such as CNNs and FNOs are not applicable in our case, as they require extensive computational resources that are not at our disposal.
> > > Furthermore, the QL baseline is the go to reduced numerical method that is being used in practice in integrated modelling [1].
> > >
> > > Our model has more parameters, but this is only natural as the problem of modelling turbulence requires more capacity than simply inferring quantities resulting from it. As demonstrated in the updated table (Weakness 1), _GyroSwin-L_ manages to capture turbulence, and on top attains a promotion on flux prediction of around 65% compared to QL and of 30% to the GPR model.
> > >
> > > That said, can you please be more specific on which other baselines you would expect to be added? This would be a very valuable input to improve our manuscript.
> > >
> > > [1] Citrin J. et al. _TORAX: A Fast and Differentiable Tokamak Transport Simulator in JAX_

---

> > > > ### Comment · Reviewer_n9nG · 2025-08-07
> > > >
> > > > Thank you for your response. To clarify my earlier point: in your main results table, the comparison baselines consist mostly of standard textbook models such as GPR, MLP, LSTM, and a basic Transformer. However, for example, in the case of Transformers, there now exist many advanced variants that have demonstrated significantly better performance. It is also well-known that Transformer-based models typically benefit from increased parameter counts, so the current comparisons may not be entirely fair.
> > > >
> > > > You mentioned that it would be helpful to know which baselines to include, so I provided just a simple example to illustrate my point. That said, I believe it is ultimately the authors' responsibility to actively identify and include fair and meaningful baselines, especially if the aim is to convincingly demonstrate the strength of a new model.

---

> > > > > ### Author Response · Authors · 2025-08-07
> > > > >
> > > > > Yes, there are more advanced transformer architectures for tabular data and sequence models that a transformer decoder, but this is partially beyond the point. If you read the rebuttal and looked at the third column it is clearly stated that __the problem solved is very different__. __GPR, MLP solve a scalar regression__ problem. __LSTM and Transformer solve a scalar time sequence__ problem. __Our model solves a 5D to 5D__ problem, it models turbulence by reconstructing the phase space, and not a single number. The $Q$ errors reported are a time-average characteristic of turbulence.
> > > > >
> > > > > The other columns beyond the flux $Q$ are only reported for quasi linear (QL) and our models, since they cannot possibly be produced by a baseline that does not process field data.

---

### Note · Authors · 2025-08-11

We appreciate the constructive feedback, recognizing both our technical contributions (accurate and efficient modeling of 5D gyrokinetic turbulence, multitask hierarchical ViT framework) and opportunities to strengthen the presentation of our work.

A major critique concerned Table 2 (reviewers n9nG, WQGA, DBSK, vuY7). We realized that our original presentation did not clearly convey the intended message: GyroSwin models turbulence in full 5D fields (2×32×8×16×85×32), whereas widely used reduced numerical models or surrogates predict only derived quantities from 5D turbulence. The baselines are established regression tools that predict the flux directly from operation parameters ([R/Lt, R/Ln, s, q] = 4-valued vector). The quasilinear model, in particular, is state-of-the-art and the most widely used reduced numerical model in practice. The task tackled by GyroSwin is several orders of magnitude more challenging, and direct comparisons may be misleading — especially since 3D and 5D phenomena are not captured by most baselines. We have clarified this in our rebuttal and made several key additions:

1. Conditioning: Added conditioning on operation parameters to the flux prediction head of GyroSwin (as in the baselines), yielding substantial gains in flux prediction.
2. Metrics: Reported Pearson correlation coefficients for diagnostics and zonal flow evaluation in Table 2, with error bars to show statistical significance.
3. Additional baselines: Included a simple Transformer model and results from scaling GyroSwin to larger parameter counts, showing significant gains.
4. Input shape clarity: Added input shape information for each method and marked n/a fields where evaluations do not apply to certain baselines.
5. Efficiency comparison: Added runtime, memory, and GFLOPs tables for methodologies applicable to 5D input, highlighting the prohibitive cost for some.
6. Ablations: Added a separate table for ablation studies of different prediction heads on a reduced data setup.

We believe these changes address the concerns raised and substantially strengthen the clarity and rigor of our work.

Thank you again for your time and effort!

Best,

The Authors

---

### Decision · Program_Chairs · 2025-09-17

**Decision:**

Accept (poster)

**Comment:**

This paper presents GyroSwin, a deep learning surrogate model for 5D gyrokinetic plasma turbulence simulations, which is a critical and computationally demanding problem in fusion research. GyroSwin extends a hierarchical Vision Transformer to 5D data and makes necessary adaptation. GyroSwin predicts the 5D distribution function, 3D electrostatic potential, and scalar fluxes, showing competitive performance against baselines.

The reviewers recognize that the paper has good motivation and clear writing, and also raise important concerns, mainly including that the results in the main Table 2 are outperformed by several baselines, the paper lacks ablation study, and lacks sufficient comparison with other methods. After the rebuttal, the authors provided an updated Table 2, where their method is now also conditioned on operational parameters (previously did not) similar to baselines. The updated Table 2 shows that the proposed GyroSwin outperforms baselines, and is the only method that can model turbulence. The authors also provide ablation study on hyperparameters. Overall the main concerns raised by the reviewers are addressed. As an application-driven paper and addresses the very important problem of fusion simulation, this paper is above the acceptance bar. The authors are required to incorporate the rebuttal improvements into the camera-ready version of the paper.